# Taxonomic revision of the genus *Amphritea* supported by genomic and *in silico* chemotaxonomic analyses, and the proposal of *Aliamphritea* gen. nov.

Ryota Yamano[1], Juanwen Yu[1], Chunqi Jiang[1], Alfabetian Harjuno Condro Haditomo[1,2], Sayaka Mino[1], Yuichi Sakai[3], Tomoo Sawabe[1]*

1 Laboratory of Microbiology, Faculty of Fisheries Sciences, Hokkaido University, Hakodate, Japan,
2 Aquaculture Department, Faculty of Fisheries and Marine Sciences, Universitas Diponegoro, Semarang, Indonesia, 3 Hakodate Fisheries Research, Hokkaido Research Organization, Local Independent Administrative Agency, Hakodate, Japan

* sawabe@fish.hokudai.ac.jp

**Data Availability Statement:** The GenBank accession number for the 16S rRNA gene

## Abstract

A Gram-staining-negative, aerobic bacterium, designated strain PT3$^T$ was isolated from laboratory-reared larvae of the Japanese sea cucumber *Apostichopus japonicus*. Phylogenetic analysis based on the 16S rRNA gene nucleotide sequences revealed that PT3$^T$ was closely related to *Amphritea ceti* RA1$^T$ (= KCTC 42154$^T$ = NBRC 110551$^T$) and *Amphritea spongicola* MEBiC05461$^T$ (= KCCM 42943$^T$ = JCM 16668$^T$) both with 98.3% sequence similarity, however, average nucleotide identity (ANI) and *in silico* DNA-DNA hybridization (*in silico* DDH) values among these three strains were below 95% and 70%, respectively, confirming the novelty of PT3$^T$. Furthermore, the average amino acid identity (AAI) values of PT3$^T$ against other *Amphritea* species were on the reported genus delineation boundary (64–67%). Multilocus sequence analysis using four protein-coding genes (*recA*, *mreB*, *rpoA*, and *topA*) further demonstrated that PT3$^T$, *Amphritea ceti* and *Amphritea spongicola* formed a monophyletic clade clearly separate from other members of the genus *Amphritea*. Three strains (PT3$^T$, *A. ceti* KCTC 42154$^T$ and *A. spongicola* JCM 16668$^T$) also showed higher similarities in their core genomes compared to those of the other *Amphritea* spp. Based on the genome-based taxonomic approach, *Aliamphritea* gen. nov. was proposed together with the reclassification of the genus *Amphritea* and *Aliamphritea ceti* comb. nov. (type strain RA1$^T$ = KCTC 42154$^T$ = NBRC 110551$^T$), *Aliamphritea spongicola* comb. nov. (type strain MEBiC05461$^T$ = KCCM 42943$^T$ = JCM 16668$^T$), and *Aliamphritea hakodatensis* sp. nov. (type strain PT3$^T$ = JCM 34607$^T$ = KCTC 82591$^T$) were suggested.

## Introduction

The genus *Amphritea*, a member of the family *Oceanospirillaceae* in the order *Oceanospirillales*, was first proposed by Gärtner et al. (2008) with the description of *Amphritea atlantica*,

sequence of the type strain is OL455018. The whole genome sequence of the strain has been deposited to DDBJ/ENA/GenBank under the accession number AP025281-AP025284, and PRJDB12633.

**Funding:** This study was supported by Kaken 19K22262. The funders had no role in study design, data collection and analysis, decision to publish, or preparation of the manuscript.

**Competing interests:** The authors have declared that no competing interests exist.

isolated from deep-sea mussels collected from a hydrothermal vent field [1]. Subsequently, six species have been proposed in this genus: *Amphritea japonica*, and *Amphritea balenae* from the sediment adjacent to sperm whale carcasses [2], *Amphritea ceti* from Beluga whale feces [3], *Amphritea spongicola* from a marine sponge [4], *Amphritea opalescens* from marine sediments [5] and *Amphritea pacifica* from a mariculture fishpond [6]. The bacteria in the genus *Amphritea* are ecophysiologically diverse, and the genus is characterized as rod-shaped, Gram-negative aerobic chemoorganotrophic, motile by means of a single polar flagellum or bi-polar flagella and catalase-positive [1, 4]. Strains in the genus also accumulate poly-β-hydroxybutyrate [1], which has been suggested as contributing to growth gaps in the sea cucumber *Apostichopus japonicus* [7]. However, no comprehensive studies on genomic characterization of the genus *Amphritea* have been undertaken.

In the process of collecting reference genomes to understand structure, function and dynamics of sea cucumber microbiome, strain PT3[T], phylogenetically unique bacterium affiliated to the genus *Amphritea*, was isolated from larvae of *Apostichopus japonicus*. Here, we report the molecular systematics of previously reported *Amphritea* species and strain PT3[T] using modern genome-based taxonomic approaches including *in silico* chemotaxonomy, and propose *Aliamphritea* gen. nov. with the reclassification of *A. ceti* and *A. spongicola* as *Aliamphritea ceti* comb. nov. and *Aliamphritea spongicola* comb. nov. and the strain PT3[T] as *Aliamphritea hakodatensis* sp. nov.

## Materials and methods

### Bacterial strains and phenotypic characterization

The strain PT3[T] was isolated from the pentactula larvae of *Apostichopus japonicus* reared in a laboratory aquarium in July 2019. Larvae were collected with 45 μm nylon mesh (FALCON Cell Strainer, Durham, USA) and ten-fold serial dilutions of the homogenate were cultured on 1/5 strength ZoBell 2216E agar plates. Bacterial colonies were purified using the same agar plate. *A. atlantica* JCM 14776[T], *A. balenae* JCM 14781[T], *A. japonica* JCM 14782[T], *A. spongicola* JCM 16668[T] and *A. ceti* KCTC 42154[T] were used as references for genomic and phenotypic comparisons against strain PT3[T]. All strains were cultured on Marine agar 2216 (BD, Franklin Lakes, New Jersey, USA). The phenotypic characteristics were determined according to previously described methods [8–11].

Cell morphology of the strain PT3[T] was observed using a transmission electron microscope JEM-1011 (JEOL, Tokyo, Japan). Cells grown in a Marine Broth 2216 (BD) at 25˚C for two days were stained with EM Stainer (Nisshin EM Co., Ltd, Tokyo, Japan) on excel-support-film 200 mesh Cu (Nisshin EM Co., Ltd, Tokyo, Japan).

Motility was observed under a microscope using cells suspended in droplets of sterilized 75% artificial seawater (ASW).

### Molecular phylogenetic analysis based on 16S rRNA gene nucleotide sequences

The nearly full length 16S rRNA gene sequence (1,404 bp) of strain PT3[T] was obtained by direct sequencing of PCR-amplified DNA. 27F and 1509R were used as amplification primers, and four primers: 27F, 800F, 920R and 1509R were used for sequencing [12]. The 16S rRNA gene nucleotide sequences of the type strains of the genus *Amphritea* and other *Oceanospirillaceae* species were retrieved from RDP (Ribosomal Database Project) [13] and NCBI databases. Sequences were aligned using Silva Incremental Aligner v1.2.11 [14]. A phylogenetic model test and maximum likelihood (ML) tree reconstruction were performed using the MEGAX

v.10.1.8 program [15, 16]. ML tree was reconstructed with 1,000 bootstrap replications using Kimura 2-parameter (K2) with gamma distribution (+G) and invariant site (+I) model. In addition, nucleotide similarities among strains were also calculated using the K2 model in MEGAX.

## Whole genome sequencing

Genomic DNA of PT3[T], *A. atlantica* JCM 14776[T], *A. japonica* JCM 14782[T], *A. spongicola* JCM 16668[T] and *A. ceti* KCTC 42154[T] was extracted from the cells grown in Marine Broth 2216 using the Wizard genomic DNA purification kit (Promega, Madison, WI, USA) according to the manufacturer's protocol. Genome sequencing was performed using both Oxford Nanopore Technology (ONT) MinION and Illumina MiSeq platforms. For the ONT sequencing, the library was prepared using Rapid Barcoding Sequence kit SQK-RBK004 (Oxford Nanopore Technologies, Oxford, UK) according to the standard protocol provided by the manufacturer. The library was loaded into flowcell (FLO-MIN 106), and a 48-hour sequencing run with Min-KNOW 3.6.0 software was performed. Basecall was performed using Guppy v4.4.1 (Oxford Nanopore Technologies). Genome sequences were also obtained from a 300 bp paired-end library prepared using the NEBNext Ultra II FS DNA Library Prep Kit for Illumina. The ONT and Illumina reads were assembled using Unicycler 0.4.8 [17]. Genomes for *A. balenae* JCM 14781[T], *A. opalescens* ANRC-JH14[T] and *A. pacifica* ZJ14W[T] were retrieved from the NCBI database, the assembly accession numbers are GCF_014646975.1, GCF_003957515.1 and GCF_016924145.1, respectively [5, 6, 18]. The whole genome sequences were annotated with DDBJ Fast Annotation and Submission Tool (DFAST) [19]. The complete genome sequences acquired in this study were deposited under AP025281-AP025284 and AP025761-AP025762 (bioproject_id PRJDB12633).

## Overall genome relatedness indices (OGRIs)

Overall genome relatedness indices (OGRIs) were calculated to determine the novelty of PT3[T]. Average nucleotide identities (ANIs) calculated using the Orthologous Average Nucleotide Identity Tool (OrthoANI) software [20] using genomes of the PT3[T] and previously described *Amphritea* type strains. *In silico* DDH values were calculated using Genome-to-Genome Distance Calculator (GGDC) 2.1 [21], results based on formula 2 was adopted, being the most robust against incomplete genomes. Average amino acid identities (AAIs) were calculated and compared between PT3[T] and other related *Oceanospirillaceae* species (S1 and S2 Tables) using an enveomics toolbox [22].

## Multilocus sequence analysis (MLSA)

MLSA was performed as previously described [8, 9]. The sequences of four protein-coding genes (*recA*, *mreB*, *rpoA*, and *topA*), essential single-copy genes in the taxa examined in this study were obtained from the genome sequences of PT3[T], *A. ceti* KCTC 42154[T], *A. spongicola* JCM 16668[T], *A. atlantica* JCM 14776[T], *A. japonica* JCM 14782[T] and other related *Oceanospirillaceae* species (S2 Table, see genome accession number in the description section below). The sequences of each gene were aligned using ClustalX 2.1 [23]. Concatenation of sequences and phylogenetic reconstruction were performed using SplitsTree 4.16.2 [24].

## Pan and core genome analysis

A total of eight genomes, including five obtained in this study (PT3[T], *A. atlantica* JCM 14776[T], *A. balenae* JCM 14781[T], *A. japonica* JCM 14782[T], *A. spongicola* JCM 16668[T] and *A. ceti* KCTC

42154[T]) and two retrieved from the NCBI database (*A. opalescens* ANRC-JH14[T] and *A. pacifica* ZJ14W[T]) were used for pangenome analysis using the program anvi'o v7 [25] based on previous studies [11, 26], with minor modifications. Briefly, contigs databases of each genome were constructed by fasta files (anvi-gen-contigs-database) and decorated with hits from HMM models (anvi-run-hmms). Subsequently functions were annotated for genes in contigs database (anvi-run-ncbi-cogs). KEGG annotation was also performed (anvi-run-kegg-kofams). The storage database was generated (anvi-gen-genomes-storage) using all contigs databases and pangenome analysis was performed (anvi-pan-genome). The results were displayed (anvi-display-pan) and adjusted manually.

## Synteny plot

To elucidate intra-species and inter-genus genome synteny among *Aliamphritea* and *Amphritea* species, synteny plot analysis was performed using in silico MolecularCloning (In Silico Biology Inc., Yokohama, Japan). A total of five complete genomic sequences determined in this study (PT3[T], *A. spongicola* JCM 1668[T], *A. ceti* KCTC 42154[T], *A. atlantica* JCM 14776[T] and *A. japonica* JCM 14782[T]) were used for this analysis. Plasmid sequences of *A. japonica* were not used for this analysis.

## *In silico* chemical taxonomy: Prediction of fatty acids, polar lipid and isoprenoid quinone using the comparative genomics approach

The genes encoding key enzymes and proteins for the synthesis of fatty acids (FAs), polar lipids and isoprenoid quinones were retrieved from the genome sequences of PT3[T] and seven previously described *Amphritea* species using *in silico* MolecularCloning ver. 7. Genomic structure and distribution of the genes were compared also using *in silico* MolecularCloning ver.7. The 3D-structure of FA desaturase encoding genes from some of the strains was predicted using Phyre2 [27].

## Results and discussion

### Molecular phylogenetic analysis based on 16S rRNA gene nucleotide sequences

Phylogenetic analysis based on 16S rRNA gene nucleotide sequences showed that strain PT3[T] was affiliated to the members of the genus *Amphritea* showing 95.4–98.3% sequence similarities, which are below the proposed threshold range for the species boundary, 98.7% [28, 29]. The strain showed high sequence similarities of 98.3% with *A. spongicola* and *A. ceti*. The maximum-likelihood tree also revealed that the PT3[T] formed a monophyletic clade with *A. spongicola* and *A. ceti* within the genus *Amphritea* (Fig 1). Even after the description of *A. pacifica* [6], two distinct lineages based on 16S gene sequences in the genus *Amphritea* have never been discussed yet, but the finding of PT3[T] showed phylogenetically more cohesion of the strain to *A. ceti* and *A. spongicola* compared to the other *Amphritea* species (Fig 1). This observation triggered further assessments of PT3[T] using molecular phylogenetic network and genomic approaches, which are frequently used in *Vibrionaceae* taxonomy [11, 30].

### Genomic features and overall genome relatedness indices (OGRIs)

Comparison of the genomic features of PT3[T] and the described *Amphritea* species showed that PT3[T], *A. spongicola* and *A. ceti* had relatively larger genome sizes (>4.9 Mb) compared to those of other *Amphritea* species (S1 Table) (see further discussion in "Pan and core genome analysis" section). The ANI values of the PT3[T] against *A. spongicola*, *A. ceti*, *A. atlantica*, *A.*

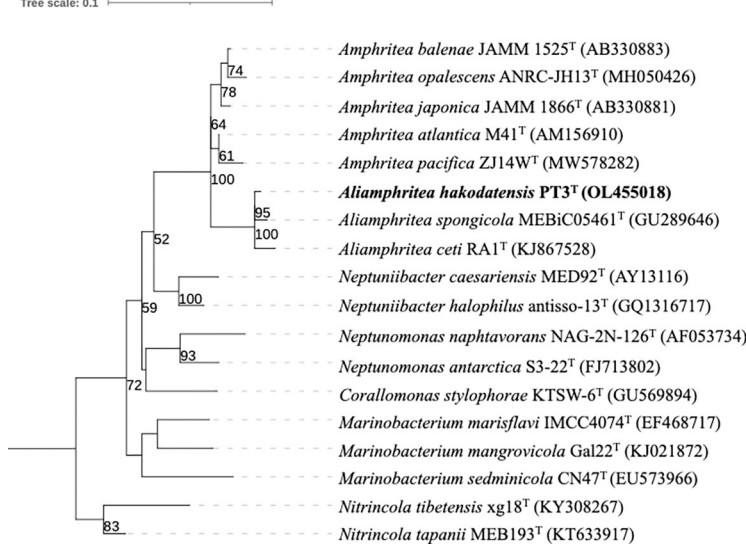

**Fig 1. A rooted ML tree based on 16S rRNA gene nucleotide sequences of strain PT3$^T$ and related type strains.** Numbers shown on branches are bootstrap values (>50%) based on 1,000 replicated analysis from maximum-likelihood algorithm. Bar, 0.1 substitutions per nucleotide position. Sequences trimmed to 1,337 bp were compared (54–1,390 position in *Al. ceti* RA1$^T$, KJ867528). *Escherichia coli* K-12 was used as an outgroup.

*pacifica*, *A. opalescens*, *A. japonica* and *A. balenae* were 89.0%, 80.1%, 72.2%, 72.2%, 71.3%, 71.7% and 72.3%, respectively (S1 Fig), which are below the species boundary threshold of 95% proposed in previous studies [31]. The *in silico* DDH values of PT3$^T$ against those species were 36.4%, 22.7%, 21.9%, 21.5%, 21.2%, 22.5% and 22.3%, respectively, and these values were also below the species delineation threshold (70%). ANI and *in silico* DDH confirmed PT3$^T$ as a novel species.

PT3$^T$ showed relatively high AAI values of 93.7% and 86.9% to *A. spongicola* and *A. ceti* (Fig 2), but these values were also below the species delineation boundary, 95–96% [32]. However, the values against other five *Amphritea* species (*A. atlantica*, *A. balenae*, *A. japonica*, *A. opalescens* and *A. pacifica*) were much lower (64.6–67.1%), and these values were on the border line for the genus delineation threshold, 65–66% [9]. The AAI values indicated that PT3$^T$, *A. spongicola* and *A. ceti* could affiliate to a novel genus.

## Multilocus sequence analysis (MLSA)

MLSA network showed that PT3$^T$, together with *A. spongicola* and *A. ceti*, form a monophyletic clade distinct from other *Amphritea* species (Fig 3). MLSA supported the proposal that those three strains should be re-classified separate from any previously described genera on the basis of phylogenetic cohesion.

## Phenotypic characterization

PT3$^T$ cells observed under a transmission electron microscope were rod-shaped (1.0–1.4 μm in length and 0.5–0.8 μm in diameter) with a single polar flagellum (S2 Fig), these morphological features were consistent with previously reported descriptions of *Amphritea* [1–6]. PT3$^T$ shared several biochemical features with *Amphritea* species, such as testing positive in oxidase, NaCl requirements for growth, ability to grow at 15˚C and 25˚C. PT3$^T$ could be distinguished from *Amphritea* spp. by a total of 43 phenotypic and biochemical features (growth at 4, 30, 37 and 40˚C, growth in 1 and 10% NaCl, catalase, indole production, hydrolysis of Tween 80,

| | Al.hakodatensis | Al.spongicola | Al.ceti | A.balenae | A.japonica | A.opalescens | A.pacifica | A.atlantica | Neptuni.caesariensis | Neptuni.marinus | Neptuni.pectenicola | Neptuno.antarctica | Neptuno.phycophila | M.aestuarii | M.georgiense | M.litorale | M.mangrovicola | Nit.tapanii | Nit.lacisaponensis | Nit.tibetensis |
|---|---|---|---|---|---|---|---|---|---|---|---|---|---|---|---|---|---|---|---|---|
| *Aliamphritea hakodatensis* | 100.0 | 93.7 | 86.9 | 67.1 | 65.8 | 65.6 | 64.8 | 64.6 | 61.0 | 59.7 | 59.5 | 59.7 | 57.8 | 58.9 | 57.7 | 57.3 | 56.4 | 57.1 | 56.0 | 55.1 |
| *Aliamphritea spongicola* | 93.7 | 100.0 | 87.0 | 67.1 | 65.8 | 65.5 | 65.0 | 64.7 | 61.1 | 60.0 | 60.0 | 59.6 | 57.7 | 59.1 | 57.8 | 57.7 | 56.8 | 57.0 | 55.9 | 55.4 |
| *Aliamphritea ceti* | 86.9 | 87.0 | 100.0 | 67.2 | 65.9 | 65.7 | 64.9 | 64.8 | 61.0 | 59.8 | 59.6 | 59.4 | 57.2 | 59.1 | 57.5 | 57.1 | 56.4 | 56.7 | 55.7 | 55.3 |
| *Amphritea balenae* | 67.1 | 67.1 | 67.2 | 100.0 | 77.6 | 75.1 | 75.4 | 75.5 | 63.6 | 61.9 | 62.2 | 62.2 | 59.4 | 60.8 | 60.0 | 59.5 | 58.3 | 58.2 | 57.2 | 56.8 |
| *Amphritea japonica* | 65.8 | 65.8 | 65.9 | 77.6 | 100.0 | 78.7 | 79.8 | 79.9 | 63.4 | 62.2 | 62.2 | 62.2 | 59.6 | 61.0 | 60.5 | 59.8 | 58.6 | 58.8 | 57.9 | 57.5 |
| *Amphritea opalescens* | 65.6 | 65.5 | 65.7 | 75.1 | 78.7 | 100.0 | 81.2 | 81.5 | 63.0 | 62.8 | 62.9 | 62.1 | 60.1 | 61.1 | 60.7 | 59.8 | 59.2 | 58.8 | 58.2 | 57.3 |
| *Amphritea pacifica* | 64.8 | 65.0 | 64.9 | 75.4 | 79.8 | 81.2 | 100.0 | 95.8 | 62.4 | 62.4 | 62.5 | 61.7 | 59.6 | 61.1 | 60.3 | 60.4 | 59.2 | 59.1 | 57.9 | 57.2 |
| *Amphritea atlantica* | 64.6 | 64.7 | 64.8 | 75.5 | 79.9 | 81.5 | 95.8 | 100.0 | 62.8 | 62.4 | 62.5 | 61.6 | 59.3 | 60.8 | 60.4 | 60.2 | 59.4 | 59.3 | 57.9 | 57.3 |
| *Neptuniibater caesariensis* | 61.0 | 61.1 | 61.0 | 63.6 | 63.4 | 63.0 | 62.4 | 62.8 | 100.0 | 73.5 | 73.2 | 61.2 | 59.8 | 60.5 | 60.5 | 59.7 | 58.6 | 59.1 | 57.9 | 57.4 |
| *Neptuniibacter marinus* | 59.7 | 60.0 | 59.8 | 61.9 | 62.2 | 62.8 | 62.4 | 62.4 | 73.5 | 100.0 | 87.7 | 60.7 | 60.0 | 60.4 | 60.4 | 59.9 | 59.0 | 59.3 | 58.2 | 57.8 |
| *Neptuniibacter pectenicola* | 59.5 | 60.0 | 59.6 | 62.2 | 62.2 | 62.9 | 62.5 | 62.5 | 73.2 | 87.7 | 100.0 | 60.9 | 60.2 | 60.3 | 59.9 | 60.0 | 59.2 | 59.7 | 58.0 | 57.7 |
| *Neptunomonas antarctica* | 59.7 | 59.6 | 59.4 | 62.2 | 62.2 | 62.1 | 61.7 | 61.6 | 61.2 | 60.7 | 60.9 | 100.0 | 66.5 | 61.0 | 59.6 | 59.6 | 59.1 | 58.6 | 57.8 | 57.4 |
| *Neptunomonas phycophila* | 57.8 | 57.7 | 57.2 | 59.4 | 59.6 | 60.1 | 59.6 | 59.3 | 59.8 | 60.0 | 60.2 | 66.5 | 100.0 | 60.2 | 59.7 | 59.9 | 59.3 | 58.5 | 57.7 | 57.3 |
| *Marinobacterium aestuarii* | 58.9 | 59.1 | 59.1 | 60.8 | 61.0 | 61.1 | 61.1 | 60.8 | 60.5 | 60.4 | 60.3 | 61.0 | 60.2 | 100.0 | 61.5 | 61.4 | 60.5 | 59.8 | 59.0 | 58.1 |
| *Marinobacterium georgiense* | 57.7 | 57.8 | 57.5 | 60.0 | 60.5 | 60.7 | 60.3 | 60.4 | 60.5 | 60.4 | 59.9 | 59.6 | 59.7 | 61.5 | 100.0 | 66.4 | 64.9 | 64.6 | 62.4 | 61.4 |
| *Marinobacterium litorale* | 57.3 | 57.7 | 57.1 | 59.5 | 59.8 | 59.8 | 60.4 | 60.2 | 59.7 | 59.9 | 60.0 | 59.6 | 59.9 | 61.4 | 66.4 | 100.0 | 74.5 | 62.8 | 61.3 | 60.6 |
| *Marinobacterium mangrovicola* | 56.4 | 56.8 | 56.4 | 58.3 | 58.6 | 59.2 | 59.2 | 60.2 | 58.6 | 59.0 | 59.2 | 59.1 | 59.3 | 60.5 | 64.9 | 74.5 | 100.0 | 62.1 | 60.5 | 59.3 |
| *Nitrincola tapanii* | 57.1 | 57.0 | 56.7 | 58.2 | 58.8 | 58.8 | 59.1 | 59.3 | 59.1 | 59.7 | 59.7 | 58.6 | 58.5 | 59.8 | 64.6 | 62.8 | 62.1 | 100.0 | 69.8 | 69.4 |
| *Nitrioncola lacisaponensis* | 56.0 | 55.9 | 55.7 | 57.2 | 57.9 | 58.2 | 57.9 | 57.9 | 57.9 | 58.2 | 58.0 | 57.8 | 57.7 | 59.0 | 62.4 | 61.3 | 60.5 | 69.8 | 100.0 | 70.7 |
| *Nitrincola tibetensis* | 55.1 | 55.4 | 55.3 | 56.8 | 57.5 | 57.3 | 57.2 | 57.3 | 57.4 | 57.8 | 57.7 | 57.4 | 57.3 | 58.1 | 61.4 | 60.6 | 59.3 | 69.4 | 70.7 | 100.0 |

**Fig 2. AAI matrix using *Aliamphritea* and related *Oceanospirillaceae*.** Reference genomes were downloaded from NCBI database (S2 Table).

antibiotics susceptibility and 25 carbon assimilation tests) (Table 1). Several traits also distinguished PT3[T] from the closely related *A. spongicola* and *A. ceti*, for example, hydrolysis of Tween 80, utilization of several organic compounds and susceptibility to SXT (Trimethoprim/Sulfamethoxazole), which indicates phenotypic cohesion in PT3[T], *A. spongicola* and *A. ceti*.

All strains showed growth at 15˚C and 25˚C and NaCl concentration of 3%, 6% and 8%. All strains tested in this study were positive for oxidase-test and hydrolysis of DNA. All strains tested in this study were negative for hydrolysis of starch, agar and gelatin, utilization of D-mannose, D-galactose, maltose, melibiose, lactose, N-acetylglucosamine, aconitate, meso-erythritol, D-mannitol, glycerol, L-tyrosine, D-sorbitol, α-ketoglutarate, xylose, trehalose, glucuronate, D-glucosamine, cellobiose, amygdalin, arabinose, D-galacturonate, glycerate, L-rhamnose, salicin, L-arginine, and L-citrulline. All strains were susceptible to gentamicin, carbenicillin (100 μg) and clarithromycin (15 μg) but not to sulfamethoxazole/trimethoprim.

## Proposal of the novel genus *Aliamphritea*

The results of the molecular phylogenetic analyses, OGRIs and classical phenotyping showed delineation of *A. ceti*, *A. spongicola* and PT3[T] from other *Amphritea* species. Here, we propose *Aliamphritea* gen. nov. with reclassification of *Amphritea ceti* and *Amphritea spongicola* as *Aliamphritea ceti* comb. nov. and *Aliamphritea spongicola* comb. nov., respectively. The strain PT3[T] is proposed as *Aliamphritea hakodatensis* sp. nov., a novel species in the genus *Aliamphritea*. Phenotypic characteristics of the genus *Aliamphritea* was compared to other *Oceanospirillaceae* phenotype results [33, 34], diagnostic feature's (morphology, flagellar arrangement and growth at 4˚C) differences from other genera were found (Table 2). To elucidate

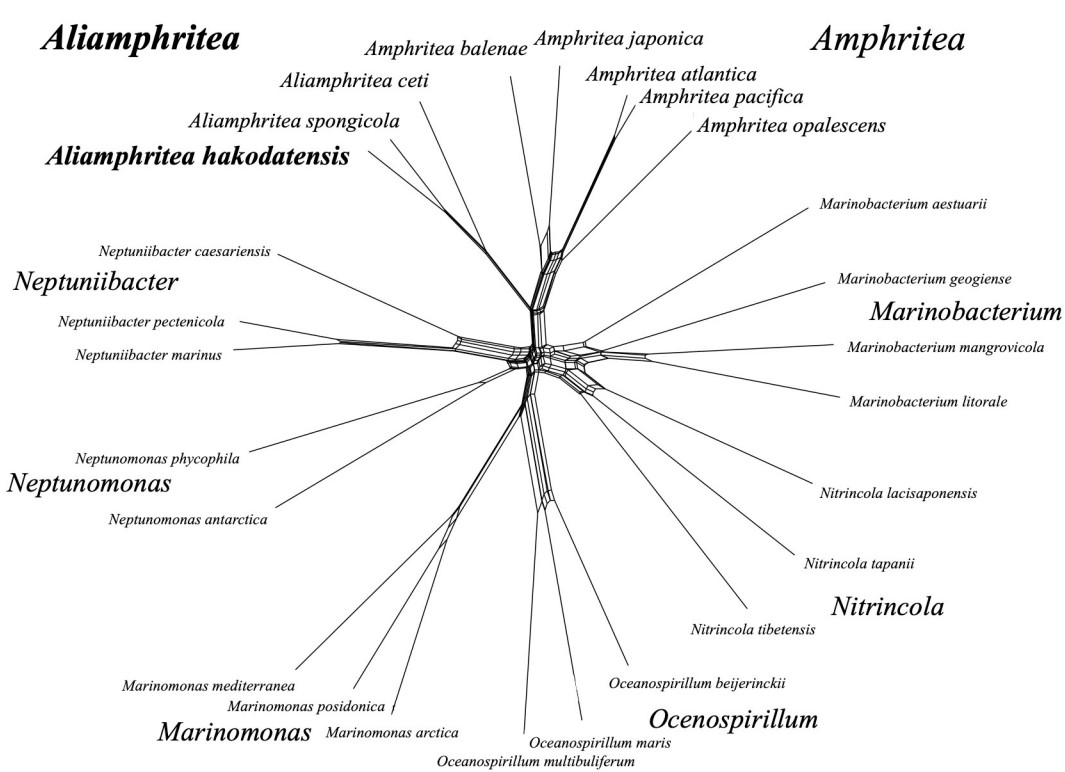

**Fig 3. MLSA network.** A list of strains and their assembly accession is provided in S2 Table.

uniqueness of genome backbone of the genus *Aliamphritea*, we further performed genome comparisons, which could also support the proposal of *Al. hakodatensis* sp. nov.

## Pan and core genome analysis

The pangenome of *Amphritea* and *Aliamphritea* species consists of 10,444 gene clusters (33,961 genes) (Fig 4). Genes were classified into *Core* for the genes present in all strains, *Aliamphritea unique* for the genes present in *Aliamphritea* species, and *Amphritea unique* for the genes present in *Amphritea* species. *Core* consisted of 1,660 gene clusters (13,930 genes). COG categories such as J (translation), ribosomal structure and biogenesis and E (amino acid transport) were abundant in this bin. *Core* also included genes encoding acetyl-CoA acetyl-transferase (*phaA*), acetoacetyl-CoA reductase (*phaB*) and poly (3-hydroxyalkanoate) polymerase subunit PhaC (*phaC*), poly (3-hydroxyalkanoate) polymerase subunit PhaE (*phaE*), completing the pathway from acetyl-CoA to poly-hydroxybutyrate. Genes coding the C4-dicarboxylate TRAP transporter system (DctMPQ), which is responsible for transportation of organic acid such as succinate, fumarate and malate were also present in this bin. Finally, DNase (exodeoxyribonuclease I,III,V,VII) and predicted lipase (phospholipase/carboxylesterase) coding genes were also present in all strains.

*Amphritea unique* consists of 446 gene clusters (2,326 genes). COG categories such as T (signal transduction mechanisms) and E (amino acid transport system) were abundant in this bin. *Amphritea unique* also included putative ABC transporter genes *yejABEF*. The transporter encoded by these genes counteracts antimicrobial peptides (AMPs) produced by animals, possibly contributing to the survival of symbiotic microbes within host environments [35]. As

**Table 1. Phenotypic characteristics of PT3[T] and related species from the genus *Amphritea*.**

| Characteristics | *Al. hakodatensi* | *Al. ceti* | *Al. spongicola* | *A. atlantica* | *A. japonica* | *A. balenae* | *A. opalescens* | *A. pacifica* |
|---|---|---|---|---|---|---|---|---|
| OF test | N | N | N | O | N | N | nd | nd |
| **Growth at** | | | | | | | | |
| 4˚C | - | - | - | + | + | + | - | - |
| 30˚C | + | + | + | + | - | - | + | + |
| 37˚C | - | - | - | + | - | - | + | + |
| 40˚C | - | - | - | + | - | - | + | - |
| **Growth in NaCl (broth)** | | | | | | | | |
| 0% | - | - | - | - | - | - | + | - |
| 1% | - | + | + | + | + | + | + | + |
| 10% | w | + | - | + | + | + | + | - |
| Catalase | + | - | w | w | + | - | + | + |
| Indole production | + | - | + | - | - | - | - | - |
| Hydrolysis of Tween 80 | + | - | + | + | - | + | nd | - |
| **Antibiotic susceptibility** | | | | | | | | |
| GM120 (Gentamicin 120) | + | +[*2] | + | + | +[*1] | +[*1] | nd | nd |
| GAT5 (Gatifloxacin 5) | + | nt | + | + | nt | nt | nd | nd |
| CTX30 (Cefotaxime 30) | + | nt | + | + | nt | nt | nd | nd |
| CB100 (Carbenicillin 100) | + | +[*2] | + | + | +[*2] | +[*2] | nd | nd |
| CLR15 (Clarithromycin 15) | + | nt | + | + | nt | nt | nd | nd |
| SXT (Sulfamethoxazole/Trimethoprim) | - | nt | + | + | nt | nt | nd | nd |
| AM10 (Ampicillin 10) | + | -[*2] | + | + | +[*1] | +[*2] | nd | nd |
| **Utilization of** | | | | | | | | |
| D-Fructose | - | - | - | + | - | - | nd | + |
| Sucrose | - | - | - | + | - | - | - | nd |
| D-Gluconate | - | - | - | + | - | - | nd | nd |
| Succinate | + | + | - | + | + | - | nd | nd |
| Fumarate | + | - | - | + | + | - | nd | nd |
| D-Mannitol | - | - | - | - | - | - | - | + |
| Citrate | + | - | - | + | - | - | + | nd |
| 4-Aminobutanoate | + | - | + | + | - | - | nd | nd |
| D-Sorbitol | - | - | - | - | - | - | - | + |
| DL-Malate | + | - | + | + | - | - | nd | nd |
| D-Glucose | - | - | - | + | - | - | - | + |
| Acetate | - | - | - | + | - | - | nd | + |
| 5-Aminovalate | - | - | + | + | - | - | nd | nd |
| Pyruvate | + | + | - | + | - | - | nd | - |
| L-Proline | + | - | - | + | - | - | nd | + |
| L-Glutamate | + | - | + | + | - | - | nd | nd |
| Putrescine | + | - | + | + | + | + | nd | nd |
| Propionate | - | - | - | + | - | - | nd | nd |
| D-Raffinose | - | - | - | + | - | - | nd | nd |
| D-Ribose | - | - | - | + | - | - | nd | nd |
| DL-Lactate | + | - | + | + | - | - | nd | nd |
| L-Alanine | + | - | - | + | - | - | nd | nd |
| L-Asparagine | - | - | - | + | - | - | nd | nd |
| Glycine | + | - | - | + | - | - | nd | - |
| L-Histidine | - | - | - | + | - | - | nd | nd |

(*Continued*)

**Table 1.** (Continued)

| Characteristics | Al. hakodatensi | Al. ceti | Al. spongicola | A. atlantica | A. japonica | A. balenae | A. opalescens | A. pacifica |
|---|---|---|---|---|---|---|---|---|
| L-Ornithine | - | - | - | + | - | - | nd | nd |
| L-Serine | + | - | - | + | - | - | nd | - |

*Amphritea opalescens* (Data from [5]); *Amphritea pacifica* (Data from [6]).

*[1]: Data from [2]

*[2]: Data from [3].

Abbreviations: In OF test, O, oxidative; N, no reaction; +, positive; -, negative; w, weakly positive; nt, not tested; nd, no data.

various associations with animal hosts have been discussed in *Amphritea*, this feature supports possible associations [1, 2, 6].

*Aliamphritea unique* consists of 1,312 gene clusters (3,999 genes). COG categories such as T (signal transduction mechanisms) and K (transcription) were abundant in this bin. One of the genes which belong to this bin encodes a tyrosine decarboxylase. This enzyme decarboxylates L-tyrosine to produce $CO_2$ and tyramine, which is an important monoamine for invertebrates which plays a similar physiological role to adrenalin for vertebrates. This suggests a possible ecological role of the *Aliamphritea* species, interacting with the nervous system of host animals through the production of tyramine [36]. In addition to the putative lipase genes distributed in the *Core*, triacylglycerol lipase genes were also observed in the *Aliamphritea unique*.

*Aliamphritea* and *Amphritea* had different pathways for metabolism of polyamines (S3 Fig). *Aliamphritea* species had two genes for putrescine biosynthesis, *speA* and *speB*. Arginine decarboxylase encoded by *speA* decarboxylates arginine into agmatine, then agmatine amidonohydrolase encoded by *speB* produces putrescine through ureohydrolysis of agmatine. *speC* encodes ornithine decarboxylase, which directly produces putrescine, this gene was only present in *Amphritea unique*. Putrescin, together with S-adenosylmethioninamine is synthesized into spermidine through putrescine aminopropyltransferase, encoded by *speE*. Nevertheless, *speE* is present in both clades, *speD*, the gene responsible for production of S-adenosylmethioninamine is only present in *Amphritea unique*. This suggests that only *Amphritea* species are potentially capable of producing spermidine independently. Finally, genes encoding putrescine—pyruvate aminotransferase (*spuC*) and 4-guanidinobutyraldehyde dehydrogenase (*kauB*) were present in both clade species. The two enzymes catalyze the conversion of putrescine to 4-aminobutanoate (GABA) through two step reactions. Putrescine, and GABA are known to be bacteria-derived amines commonly found in gastro-intestinal environment [37]. In particular, GABA is a neurotransmitter widely distributed in animals including sea cucumber [38], which indicates that *Aliamphritea* species might influence the nervous system of their host during the developmental stage [39]. Finally, putative tricarboxylic transport membrane protein (TctABC) coding genes, which is responsible for the transportation of lactate, pyruvate and citrate were found in all strains except for *A. japonica*.

Pan-genome analyses also revealed that genes from two COG categories, K (Transcription) and E (Amino acid transport and metabolism functions) were more abundant in *Aliamphritea* species than those found in *Amphritea* species, which could contribute to the genome expansion in *Aliamphritea* species (S1 Table). In particular, number of genes (122 in average) responsible to LysR-type transcription regulators (LTTRs), which occupied 30% of genes categorized in K, was significantly higher ($P<0.01$, Welch $t$-test) in *Aliamphritea* than that (65 in average) in *Amphritea*. LTTRs are DNA-binding protein transcriptional regulators which are involved in diverse functions including metabolism, quorum sensing, virulence and motility, commonly regulating a single divergently transcribed gene [40]. In-depth genome BLAST

**Table 2. Phenotypic characteristics of *Aliamphritea* and *Amphritea* and genus level comparison within the family *Oceanospirillaceae*.**

| | *Aliamphritea* | *Amphritea* | *Marinomonas* | *Oceanospirillum* | *Neptuniibacter* | *Neptunomonas* | *Nitrincola* | *Marinobacterium* | *Pontibacterium* |
|---|---|---|---|---|---|---|---|---|---|
| Morphology | Rods or ovoids | Rods | Helical, curved or straight rods | Helical | Rods | Rods | Rods | Rods | Rods |
| Number and arrangement of flagella | 1 polar or none | 1 polar or bipolar tufts | 1 polar or bipolar tufts | bipolar tufts | nd | 1 polar | 1 polar | 1 polar | 1 polar |
| Optimal temperature (°C) | 25–30 | 20–30 | 4–40 | 25–32 | 15–37 | nd | 37 | 37 | 30 |
| Growth at 4°C | - | + | d | d | - | + | - | + | - |
| Growth at 45°C | - | - | - | - | - | - | nd | - | - |
| Optimal NaCl (%) for growth | 2.0–2.5 | 2.0–3.0 | nd | 0.5–8.0 | nd | 1.75–7.0 | 5 | 0.6–2.9 | 2.0 |
| Maximal NaCl (%) for growth | 8.0 | 6.0 | nd | 8.0 | 6.0 | 7.0 | 8.0 | 11.7 | 6.0 |
| Nitrate reduction to nitrite | + | + | d | - | - | - | nd | - | + |
| Oxidase | + | + | d | + | + | + | + | - | - |
| Catalase | d | + | d | d | + | + | + | + | + |
| Gelatin liquefaction | - | d | d | d | - | - | - | nd | - |
| Starch hydrolysis | - | nd | d | - | - | - | - | nd | - |
| Utilization of | | | | | | | | | |
| D-Glucose | - | d | + | - | - | + | - | + | + |
| D-Fructose | - | d | d | - | - | + | - | + | nd |
| D-Mannose | - | - | d | - | - | - | - | + | - |
| Sucrose | - | d | d | - | - | nd | - | - | nd |
| Cellobiose | - | - | d | - | - | nd | - | + | nd |
| D-Mannitol | - | d | d | - | - | + | - | + | - |
| Glycerol | - | - | d | - | - | + | - | + | nd |
| Gluconate | - | d | d | nd | - | - | nd | nd | - |
| L-Arginine | - | - | d | - | + | nd | - | - | - |
| Acetate | - | d | d | d | + | + | + | + | nd |
| PHB accumulation | nd | + | - | + | + | + | nd | - | + |
| Mol% G + C in DNA | 47–52 | 48–51 | 41–50 | 45–50 | 47 | 46 | 47.4 | 54.9 | 51.5 |
| Major ubiquinone | Q-8 | Q-8 | Q-8 | Q-8 | Q-8 | Q-8 | nd | Q-8 | Q-7, Q-8 |
| Type species | *Al. ceti* | *A. atlantica* | *M. communis* | *O. linum* | *N. caesariensis* | *N. naphthovorans* | *N. lacisaponensis* | *M. georgiense* | *P. granulatum* |

Original data from [33, 34]. Phenotypic data for *Aliamphritea* and *Amphritea* are from this study and [3–6].

+, present in all strains; -, lack in all strains; d, differs among strains; nd, not determined; W, weak reaction.

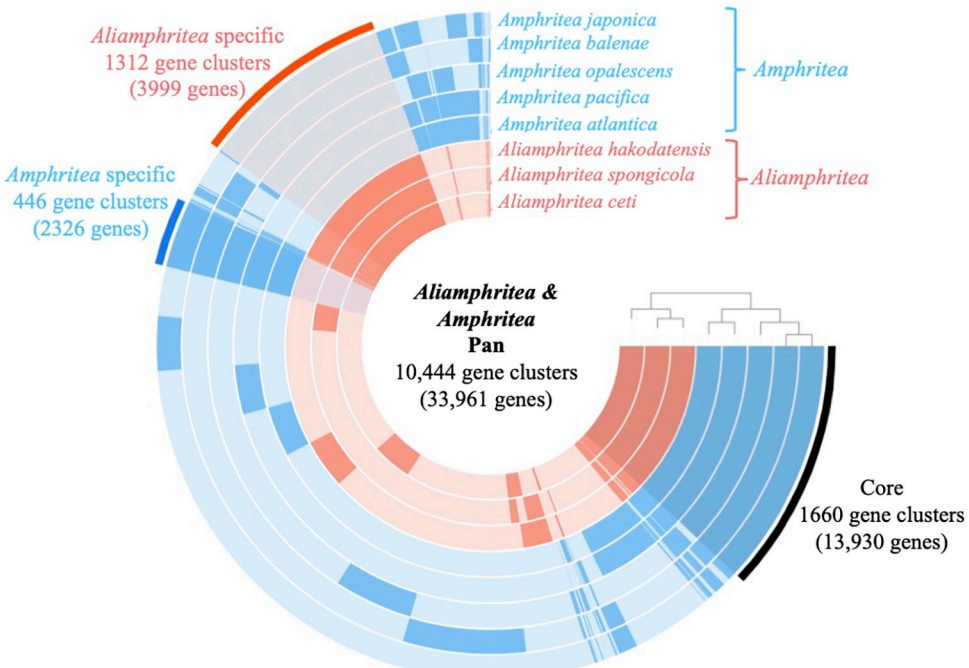

**Fig 4. Anvi'o representation of the pangenome of the *Aliamphritea* and *Amphritea* species.** Layers represent each genome, and the bars represent the occurrence of gene clusters. The darker colored areas of the bars belong to one of the three bins: *Core*, *Aliamphritea unique* or *Amphritea unique*.

comparison did not find in/del of proper gene clusters and/or regions among *Aliamphritea* and *Amphritea* genomes, also supporting the idea that the major causes of genome expansion in *Aliamphritea* were LTTRs. In the aspect of ecophysiology of *Aliamphritea* spp., they were isolated from feces of Beluga whale, sea sponge, and sea cucumber larvae [3, 4], which are likely to constitute active heterotrophic microbial communities less affected by nutritional limitations. As reported in the *Pelagibacter* genomes, the rather low number of transcriptional regulators in the genome is related to fewer transcribed protein-coding genes due to nutritional limitations and/or availabilities [41], therefore, fewer-limitations of nutrients for *Aliamphritea* in their habitat compared to *Amphritea* spp., when most of them are isolated from cold extreme environments [1, 5], might drive the genome expansions acquiring LTTRs.

Core- and pan-genome analyses with phenotypic features described below also revealed ecological features of *Amphritea* and *Aliamphritea* species as byproduct users, utilizing metabolites from another microbe [42]. Genes encoding transporters of organic acids such as citrate, succinate, fumarate, lactate, malate, pyruvate were widely distributed in both clades, suggesting the ecophysiology of strains in those genera. In addition, the absence of major genes responsible for polysaccharide degrading enzymes and the presence of lipase and DNase genes in *Amphritea* and *Aliamphritea* genomes suggest effective strategies using lipid and/or nucleic acids surviving in natural assemblages [42]. This is consistent with animal associated ecological features of the *Amphritea* and *Aliamphritea* species [1–4], and *Al. hakodatensis* is also likely to be a member of the *Apostichopus japonicus* larval microbial consortium.

Possible protease genes maintaining cellular homeostasis such as ATP-dependent protease ClpP were present in the *Core* bin, but absence of apparent extra-cellular protease genes in the *Core* also supports the idea that *Amphritea* and *Aliamphritea* are byproduct users, who are unlikely to be more efficient consumers of protein and the related peptides [42].

## Synteny plot

Synteny plot clearly demonstrates similar gene arrangements among *Aliamphritea* species while showing less similarity to *Amphritea* species (Fig 5), supporting genomic cohesion of the novel genus. Furthermore, genome comparisons between *Al. hakodatensis*, *A. atlantica* and *A. japonica* (Fig 5C and 5D) indicate an inversion event which likely occurred during the divergence of *Aliamphritea* and *Amphritea*. Comparison between the two *Amphritea* species, *A. atlantica* and *A. japonica* also indicated an inversion (Fig 5E). These results implicate multiple genomic inversions, which may be responsible for the divergence of *Amphritea* and *Aliamphritea* (S4 Fig).

## *In silico* chemical taxonomy

Fatty acids, polar lipids and isoprenoid quinones are common subjects for chemotaxonomic analyses. We performed *in silico* chemotaxonomy among *Amphritea* and *Aliamphritea* species based on comparative genomic approach, as an alternative to the more traditional chemotaxonomy (S3 Table).

Reported cellular fatty acids of *Amphritea* and *Aliamphritea* species are mainly linear, mono-unsaturated or saturated consisted of C16:0, C16:1 and C18:1, with small amount of C10:0 3-OH (S3 Table) [1–6]. Pangenomic analysis among described *Amphritea* species reconstructed the basic type II fatty acid biosynthesis (FASII) pathway driven by FabABFDGIVZ and AccABCD, which is very similar to that of *E. coli* [43] (Fig 6 and Table 3). The FASII pathway could contribute to three major FAs (C16:0, C16:1 and C18:1) of both genera. In particular, long-chain saturated fatty acid (C16:0) is one of the main features of *Amphritea* and *Aliamphritea* fatty acids, comprising approximately 10–30% of the total (S3 Table). C16:0 is one of the main products of the FASII pathway, meaning PT3[T] could produce C16:0 (Fig 6). Mono-unsaturated fatty acids are also major features of the fatty acid profile. C16:1 and C18:1 together make up over 60% of the total fatty acids (S3 Table). Monounsaturated fatty acids can be produced through ω7 mono-unsaturated fatty acid synthesis initiated by isomerization of trans-2-decenoyl-ACP into cis-3-decenoyl-ACP by FabA (Fig 6). After elongation by FabB, the acyl chain is returned to the FASII pathway and goes through further elongation, producing C16:1ω7c and C18:1ω7c [44]. All strains, including *Al. hakodatensis* strain PT3[T] have *fabA* and *fabB*, thus it is suggested that this species is also capable of producing C16:1ω7c and C18:1ω7c. Furthermore, 3-hydroxylated FAs, which are the primary fatty acids in lipid A as well as in ornithine-containing lipids, could be supplied by the FASII pathway since 3-hydroxy-acyl-ACP is known to be normally intermediated in the FAS II elongation cycle [44]. Since no genes responsible for the synthesis of ornithine-containing lipids were found in *Al. hakodatensis* or any other species in either genus, it is likely that 3-hydroxylated FAs in these species originate in lipid A.

Fatty acid desaturase (Des) can produce unsaturated fatty acid as an alternative to the FASII system. While unsaturated fatty acid synthesis by FabA occurs in anaerobic conditions, fatty acid desaturases are known to function in aerobic conditions [45]. Three fatty acid desaturase homologs (*Des1-3*) were found in the genome of *Al. hakodatensis*, *Al. ceti* and *Al. spongicola*, while only one homolog, *Des4* was found in *A. japonica* and *A. balenae* (Table 3). 3D-structure prediction using Phyre2 program shows that these enzymes are likely to be stearoyl-CoA desaturase (SCD), with >99.8% confidence score to the template sequence (S4 Table). Amino acid alignment of Des1-4 also reveals the presence of histidine clusters (HXXXH, HXXHH), which are essential for enzyme activity (S5 Fig) [46]. SCD introduces *cis* double bond at the Δ9 position of palmitoyl-CoA (C16:0) or stearoyl-CoA (C18:0), producing palmitoleoyl-CoA (C16:1 ω7c) or oleoyl-CoA (C18:1 ω9c). While palmitoleic acid (C16:1 ω7c) is ubiquitous in

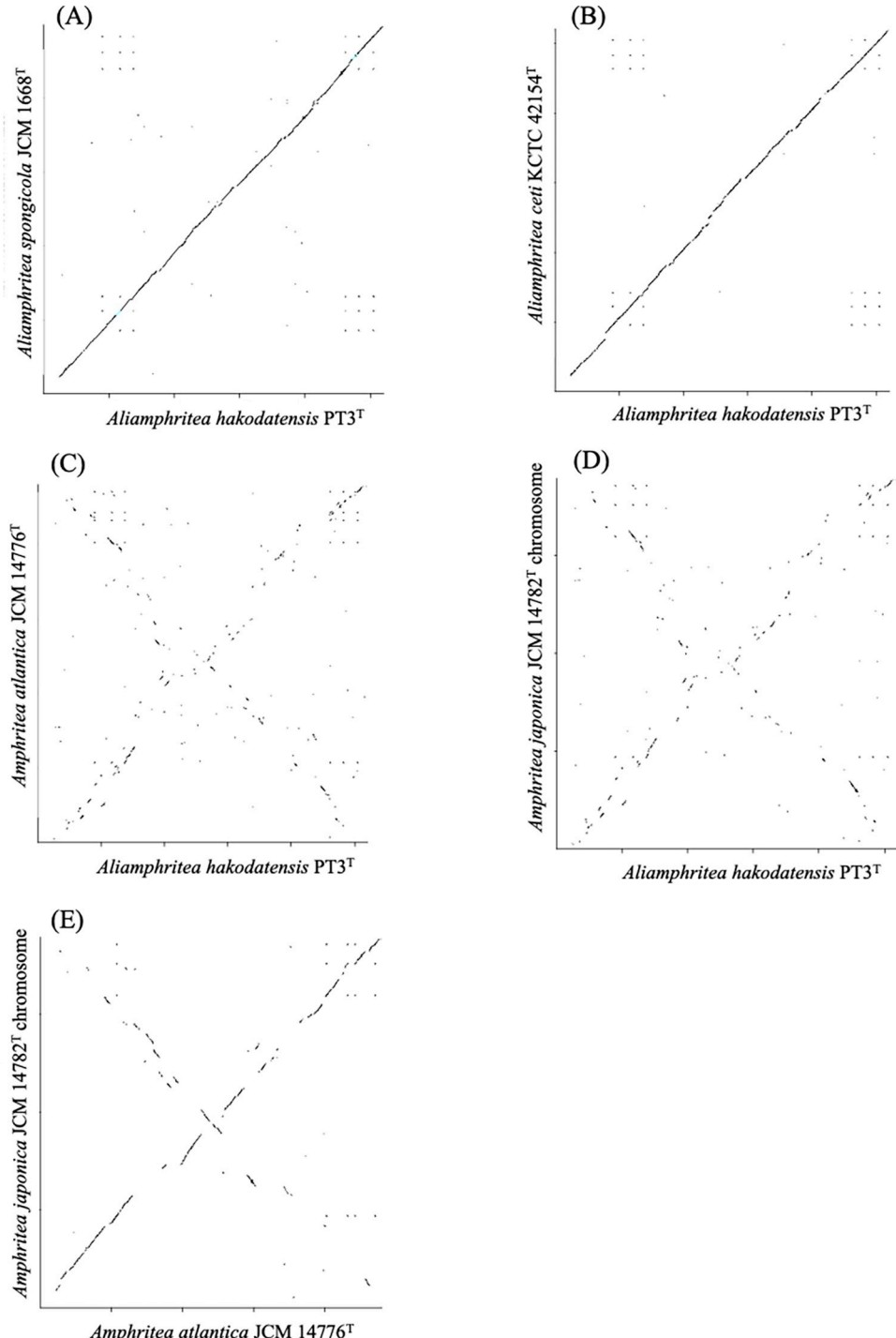

**Fig 5. Syntenic dotplot comparison of *Aliamphritea* and *Amphritea* type strains.** Dots closer to the diagonal line represents collinear arrangement between two homologous genes in two genomes. (A), (B) Intra-genus comparison of *Aliamphritea* species. (C), (D) Inter-genus comparison between *Amphritea* and *Aliamphritea* species. (E) Intra-genus comparison of *Amphritea* species.

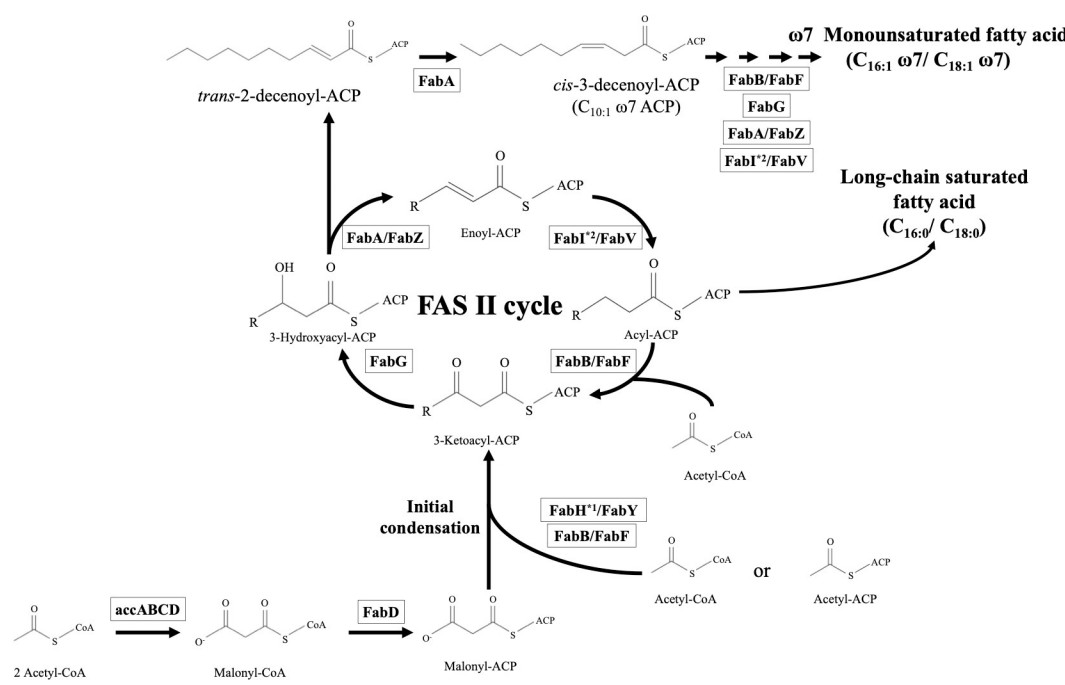

**Fig 6. Predicted fatty acid synthetic pathway in *Amphritea* and *Aliamphritea* species.** Genes encoding each enzyme were present in all strains unless stated otherwise. [*1]: Only present in *Al. hakodatensis*, *Al. ceti* and *Al. spongicola*. [*2]: Only present in *Al. ceti*, *A. atlantica*, *A. japonica*, *A. balenae* and *A. pacifica*. ACP: acyl-carrier protein; AccABCD: acetyl-CoA carboxylase complex; FabD: malonyl-CoA: ACP transacylase; FabH/FabY: 3-ketoacyl-ACP synthase III; FabB: 3-ketoacyl-ACP synthase I; FabF: 3-ketoacyl-ACP synthase II; FabG: 3-ketoacyl-ACP reductase; FabA: 3-hydroxyacyl-ACP dehydratase/trans-2-decenoyl-ACP isomerase; enoyl-ACP reductase; FabZ: 3-hydroxyacyl-ACP dehydratase.

**Table 3. FAS associated genes composition of the eight strains.**

|  | *Al. hakodatensis* | *Al. ceti* | *Al. spongicola* | *A. atlantica* | *A. japonica* | *A. balenae* | *A. opalescens* | *A. pacifica* |
|---|---|---|---|---|---|---|---|---|
| *accA* | + | + | + | + | + | + | + | + |
| *accB* | + | + | + | + | + | + | + | + |
| *accC* | + | + | + | + | + | + | + | + |
| *accD* | + | + | + | + | + | + | + | + |
| *fabD* | + | + | + | + | + | + | + | + |
| *fabH* | + | + | + | - | - | - | - | - |
| *fabY* | + | + | + | + | + | + | + | + |
| *fabB* | + | + | + | + | + | + | + | + |
| *fabF* | + | + | + | + | + | + | + | + |
| *fabG* | + | + | + | + | + | + | + | + |
| *fabA* | + | + | + | + | + | + | + | + |
| *fabZ* | + | + | + | + | + | + | + | + |
| *fabI* | - | - | + | + | + | + | - | + |
| *fabV* | + | + | + | + | + | + | + | + |
| *Des1* | + | + | + | - | - | - | - | - |
| *Des2* | + | + | + | - | - | - | - | - |
| *Des3* | + | + | + | - | - | - | - | - |
| *Des4* | - | - | - | + | - | - | - | + |

+: genes presence; -: absence.

*Amphritea* cellular fatty acid profiles, there is no record of oleic acid (C18:1 ω9c) in previous chemotaxonomic properties [1–6]. This suggests that Des1-4 is responsible for the production of palmitoleic acid (C16:1 ω7c) in aerobic environments and is unlikely to be involved in synthesis of other unsaturated fatty acids that are unconfirmed in previously described *Amphritea* and *Aliamphritea* species, such as polyunsaturated fatty acid (PUFA). In addition to this, PUFA synthase complex consisting of *pfa* genes and *ole* genes were not found in any of the eight strains [47].

Branched-chain fatty acid is also unlikely to be produced. Branched-chain fatty acid synthesis requires 3-ketoacyl-ACP synthase III which accepts branched chain primers. The *Al. hakodatensis* strain has the same 3-ketoacyl-ACP synthase III genes (*fabY*, *fabH*) as *Al. spongicola* and *Al. ceti* (Table 3), which have a fatty acid profile which is mostly linear (S3 Table), thus *Al. hakodatensis* is also unlikely to produce branched-chain fatty acids.

Comparative genome survey of the genes responsible for the FAS II pathway on the *Al. hakodatensis* genome reveals the presence of a core gene set, which is mostly similar to the other seven strains (Fig 6, Table 3). Synteny and genomic structures of FAS II core genes are likely to be retained between described *Amphritea* and *Aliamphritea* species (S6 and S7 Figs), which could lead to the conclusion that the novel strain is capable of producing similar FA profiles, mainly consisting of C16:0, C16:1ω7c, and C18:1ω7c. *Al. hakodatensis* is also potentially capable of making C10:0 3-OH which is commonly found in both genera because of the presence of the LpxA gene. LpxA is responsible for the incorporation of 3-hydroxyacyl to UDP-N-acetyl-alpha-D-glucosamine, which is a primary reaction to the biosynthesis of lipid A [48].

Pangenomic analysis among *Amphritea* and *Aliamphritea* spp. also revealed a complete gene set for the production of PG and PE; *plsX*, *plsY*, *plsC*, *cdsA*, *pssA*, *psd*, *pgsA* and *pgpA* (S5 Table). Furthermore, *cls*, which is responsible for the production of DPG [49], was detected in the genomes of four strains, *A. atlantica*, *A. balenae*, *A. opalescens* and *A. pacifica*. This suggests that while *A. opalescens* is the only *Amphritea* species with DPG in its polar lipid profile (S5 Table) [5], *A. atlantica*, *A. balenae* and *A. pacifica* are also potential DPG producers. Comparative genomics using five complete genomes reveal that *Al. hakodatensis* possesses gene sets for PG and PE production as polar lipids, showing particularly high similar gene synteny with *Al. spongicola* and *Al. ceti* (S8 Fig).

The only respiratory quinone reported from previously described *Amphritea* and *Aliamphritea* species is ubiquinone-8 (Q-8) (S3 Table). Biosynthesis of Q-8 consists of nine steps, and Ubi proteins are involved in each reaction (S9 Fig) [50]. The side chain of Q-8 consists of eight isoprene units, originating in the side-chain precursor octaprenyl-diphosphate, which is synthesized by IspAB (S9 Fig). Core genes include *ubi* genes responsible for the biosynthetic pathway (*ubiC*, *ubiA*, *ubiD*, *ubiX*, *ubiI*, *ubiG*, *ubiH*, *ubiE*), and *ispAB*. In addition to those, core genes include three genes (*ubiB*, *ubiJ*, *ubiK*) coding accessory proteins also required for Q-8 biosynthesis, but with rather hypothetical functions. UbiB is thought to be responsible for the extraction of ubiquinone precursors from the membrane while UbiJ and UbiK is thought to introduce ubiquinone intermediates to Ubi enzymes such as UbiIGHEG [50]. Ubi genes of *Al. hakodatensis* share a similar gene structure with the other seven strains, showing especially high similarities with other *Aliamphritea* members (S10 Fig). The distribution of ubiquinone-associated genes of five strains with complete genomes also shows that *Al. hakodatensis* has similar gene distribution to *Al. ceti* and *Al. spongicola* (S11 Fig), which leads to the suggestion that the predominant ubiquinone of *Al. hakodatensis* is Q-8.

Recently, further development of novel chemotaxonomy tools based on genome information was suggested in the description of bacterial species as an alternative to classical experimental chemotaxonomy [51]. The *in silico* chemotaxonomy approach is one way to achieve this, and this methodology has been used exclusively in *Corynebacterium* and *Turicella* [52].

We also applied the *in silico* chemotaxonomy approach in this study to *Amphritea* and *Aliamphritea* in the class *Gammaproteobacteria*, to estimate chemotaxonomic features such as fatty acid profiling, respiratory quinone type, and polar lipid profiling. Using complete genome sequences, we can easily predict the backbone of chemotaxonomic features of strains of interest by evaluating the presence/absence of the genes associated with biosynthetic pathways. By comparing these to the *E. coli* genome, we can find the *in silico* chemotaxonomy is capable of being applied to bacteria belonging to the class *Gammaproteobacteria*, which means *in silico* chemotaxonomy could be used in a wide range of bacterial taxa/species, in which the complete genomes have already been determined. This approach is also effective in predicting the chemotaxonomic features of less common bacterial strains, of which enough bacterial cell mass for chemotaxonomic experiments is unlikely to be collected. However, there are still difficulties in estimating factors regulating fatty acid chain length, and quantitative amounts of fatty acids. Further biochemistry and structural prediction of enzymes and/or proteins responsible to fatty acid synthesis is also needed.

## Conclusions

Using the results of modern genome taxonomic study combined with classical phenotyping, which fulfills phylogenetic, genomic, and phenotypic cohesions, we propose *Aliamphritea* gen. nov. with reclassification of *Amphritea ceti* RA1[T] and *Amphritea spongicola* MEBiC05461[T] as *Aliamphritea ceti* comb. nov. (RA1[T] = KCTC 42154[T] = NBRC 110551[T]) and *Aliamphritea spongicola* comb. nov. (MEBiC05461[T] = JCM 16668[T] = KCCM 42943[T]), respectively. The strain PT3[T] represents a novel species in the genus *Aliamphritea*, for which the name *Aliamphritea hakodatensis* sp. nov. (PT3[T] = JCM 34607[T] = KCTC 82591[T]) is proposed.

### Description of *Aliamphritea* gen. nov

*Aliamphritea* (A.li. am.phri'tea. L. masc. pron. *alinus*, other, another; N.L. fem. n. *Amphritea*, a name of a bacterial genus; N.L. fem. n. *Aliamaphritea*, the other *Amphritea*).

Members are mesotrophic, Gram-negative rods or ovoid belonging to the class *Gammaproteobacteria*. Shows growth at 15˚C, 25˚C, and 30˚C, with 3.0–8.0% (w/v) NaCl. Positive for oxidase. Members of this genus have a typical FAS II pathway gene set. Members also have complete synthetic pathways for the production of phosphatidylglycerol, phosphatidylethanolamine and ubiquinone-8. All members were isolated from marine animals related sources such as Beluga whale feces, marine sponge, and *Apostichopus japonicus* larvae. DNA G+C content is 47.1–52.2%. The range of estimated genome sizes based on the complete genome sequences is 5.0 Mb to 5.2 Mb. Type species is *Aliamphritea ceti*.

### Description of *Aliamphritea ceti* comb. Nov

*Aliamphritea ceti* (ce'ti. L. gen. n. *ceti*, of a whale). Basonym: *Amphritea ceti* Kim et al. 2014 [6].

The description is the same as that published for *Amphritea ceti* by Kim et al. (2014) [6]. The type strain is RA1[T] = KCTC 42154[T] = NBRC 110551[T]. The complete genome nucleotide sequence is deposited in the DDBJ/ENA/GenBank under the accession number AP025282 (PRJDB12633).

### Description of *Aliamphritea spongicola* comb. Nov

*Aliamphritea spongicola* (spon.gi'co.la. L. fem. n. *spongia*, a sponge; L. masc./fem. suffix n. -*cola* (from L. masc./fem. n. *incola*), inhabitant; N.L. n. *spongicola*, inhabitant of sponges). Basonym: *Amphritea spongicola* Jang et al. 2015 [4].

The description is the same as that published for *Amphritea spongicola* by Jang et al. (2015) [4]. The type strain is MEBiC05461[T] = KCCM 42943[T] = JCM 16668[T]. The complete genome nucleotide sequence is deposited in the DDBJ/ENA/GenBank under the accession number AP025283 (PRJDB12633).

## Description of *Aliamphritea hakodatensis* sp. Nov

*Aliamphritea hakodatensis* sp. nov. (ha.ko.da.ten'sis. N.L. fem. adj. *hakodatensis*, from Hakodate, referring to the isolation site of the strain).

Gram-negative, motile with single polar flagellum. Cells are rod-shaped, 1.0–1.4 μm in length and 0.5–0.8 μm in diameter. Colonies on MA are cream and 0.5–0.75 mm in diameter after culture for 3 days. No pigmentation and bioluminescence are observed. The DNA G+C content is 52.2% and genome size is 5.21 Mb. Growth occurs at 15°C, 25°C and 30°C, with NaCl concentration of 3%, 6%, 8%, 10%. Susceptible for ampicillin (10 μg), cefotaxime (30 μg), gatifloxacin (5 μg), Positive for oxidase- and catalase-test, production of indole, nitrate reduction, hydrolysis of tween 80 and DNA, utilization of succinate, fumarate, citrate, g-aminobutyrate, DL-malate, pyruvate, L-proline, L-glutamate, putrescine, DL-lactate, L-alanine, glycine and L-serine. Negative for hydrolysis of starch, agar and gelatin, utilization of D-mannose, D-galactose, D-fructose, sucrose, maltose, melibiose, lactose, D-gluconate, N-acetylglucosamine, aconitate, meso-erythritol, D-mannitol, glycerol, L-tyrosine, D-sorbitol, α-ketoglutarate, xylose, D-glucose, trehalose, glucuronate, acetate, D-glucosamine, δ-aminovalate, cellobiose, propionate, amygdalin, arabinose, D-galacturonate, glycerate, D-raffinose, L-rhamnose, D-ribose, salicin, L-arginine, L-asparagine, L-citrulline, L-histidine and L-ornithine.

The type strain PT3[T] (JCM 34607[T] = KCTC 82591[T]) was isolated from a pentactula larvae of *Apostichopus japonicus* reared in a laboratory aquarium in Hokkaido University, Hakodate, Japan. The GenBank accession number for the 16S rRNA gene sequence of the type strain is OL455018. The complete genome sequence of the strain is deposited in the DDBJ/ENA/GenBank under the accession number AP025281 (PRJDB12633).

## Supporting information

**S1 Fig. Heat map representation of ANI values of *Aliamphritea* and *Amphritea* species.**
(PDF)

**S2 Fig. An electron micrograph of negatively stained *Aliamphritea hakodatensis* PT3[T] cell.**
The bar represents 1 μm.
(PDF)

**S3 Fig. Predicted polyamine metabolism pathways in *Aliamphritea* and *Amphritea* species.**
(PDF)

**S4 Fig. Evolutionary history of *Aliamphritea* and *Amphritea* genome arrangement.**
(PDF)

**S5 Fig. Amino acid sequence alignment of *Des*1-4.**
(PDF)

**S6 Fig. Genomic distribution of *fab* and associated genes.** Protein/enzyme name each gene is coding: *fabA*: 3-hydroxyacyl-ACP dehydrase/trans-2-decenoyl-ACP isomerase; *fabB*: 3-ketoacyl-ACP synthase I; *fabD*: malonyl-CoA: ACP transacylase; *fabF*: 3-ketoacyl-ACP synthase II; *fabG*: 3-ketoacyl-ACP reductase; *fabH*: 3-ketoacyl-ACP synthase III; *fabI*: enoyl-ACP reductase I; acetyl-CoA; *fabV* enoyl-ACP reductase; *fabY*: 3-ketoacyl-ACP synthase; *fabZ*: 3-hydroxyacyl-ACP dehydratase; *accABCD*: carboxylase complex; *plsX*: phosphate

acyltransferase; *acpP*: Acyl-carrier protein.
(TIF)

**S7 Fig. Genomic structure of *Amphritea* and *Aliamphritea fab* and associated genes.**
(PDF)

**S8 Fig. Genomic distribution of genes associated with PG, PE and DPG production.** *plsC*:
1-acyl-sn-glycerol-3-phosphate acyltransferase; *plsX*: phosphate acyltransferase; *plsY*: acyl
phosphate: glycerol-3-phosphate acyltransferase *cdsA*: phosphatidate cytidylyltransferase;
*pssA*: CDP-diacylglycerol—serine O-phosphatidyl transferase; *psd*: phosphatidylserine decar-
boxylase; *pgsA*: CDP-diacylglycerol-glycerol-3-phosphate 3-phosphatidyltransferase; *pgpA*:
phosphatidyl glycerophosphatase A; *cls*: cardiolipin synthase.
(TIF)

**S9 Fig. Predicted Q-8 synthetic pathway in *Amphritea* and *Aliamphritea* species.**
(PDF)

**S10 Fig. Genomic structure of *ubi* and associated genes.**
(PDF)

**S11 Fig. Genomic distribution of ubiquinone associated genes.** *ubiA*: 4-hydroxybenzoate poly-
prenyltransferase; *ubiB*: protein kinase; *ubiC*: chorismite lyase; *ubiD*: 4-hydroxy-3-polyprenyl-
benzoate decarboxylase; *ubiE*: dimethylmenaquinone methyltransferase / 2-methoxy-
6-polyprenyl-1,4-benzoquinol methylase; *ubiF*: 3-demethoxyubiquinol 3-hydroxylase; *ubiG*:
2-polyprenyl-6-hydroxyphenyl methylase / 3-demethylubiquinone-9 3-methyltransferase; *ubiH*:
2-octaprenyl-6-methoxyphenol hydroxylase; *ubiI*: 2-polyprenylphenol 6-hydroxylase; *ubiJ*: ubi-
quinone biosynthesis accessory factor; *ubiK*: ubiquinone biosynthesis accessory factor; *ubiX*: flavin
prenyltransferase; *ispA*: farnesyl diphosphate synthase; *ispB*: octaprenyl-diphosphate synthase.
(TIF)

**S1 Table. Genome properties of *Aliamphritea* and *Amphritea* species.**
(PDF)

**S2 Table. List of other *Oceanospirillaceae* genomes used for genome taxonomy of *Aliam-
phritea* and *Amphritea*.** +: used, -: not used.
(PDF)

**S3 Table. Fatty acid, isoprenoid quinone and polar lipid profile of previously reported
*Amphritea* and *Aliamphritea* species.** PG, phosphatidylglycerol; PE, phosphatidylethanol-
amine; DPG, diphosphatidylglycerol; GPL: glycophospholipid. nd: not determined.
(PDF)

**S4 Table. Results of 3D-structure prediction of Des1-4 by Phyre2.**
(PDF)

**S5 Table. PG, PE and DPG associated genes composition of each strain.** +: genes presence;
-: absence.
(PDF)

## Acknowledgments

We gratefully thank for Professor (Emeritus) Aharon Oren, The Hebrew University of Jerusa-
lem, for his advice on bacterial names.

## Author Contributions

**Conceptualization:** Sayaka Mino, Yuichi Sakai, Tomoo Sawabe.

**Data curation:** Juanwen Yu, Chunqi Jiang, Alfabetian Harjuno Condro Haditomo, Tomoo Sawabe.

**Formal analysis:** Ryota Yamano, Juanwen Yu, Sayaka Mino, Tomoo Sawabe.

**Funding acquisition:** Sayaka Mino, Yuichi Sakai, Tomoo Sawabe.

**Investigation:** Tomoo Sawabe.

**Methodology:** Juanwen Yu, Chunqi Jiang, Alfabetian Harjuno Condro Haditomo, Sayaka Mino, Tomoo Sawabe.

**Project administration:** Sayaka Mino, Yuichi Sakai, Tomoo Sawabe.

**Resources:** Tomoo Sawabe.

**Software:** Tomoo Sawabe.

**Supervision:** Sayaka Mino, Yuichi Sakai, Tomoo Sawabe.

**Validation:** Tomoo Sawabe.

**Visualization:** Tomoo Sawabe.

**Writing – original draft:** Ryota Yamano, Juanwen Yu, Sayaka Mino, Yuichi Sakai, Tomoo Sawabe.

**Writing – review & editing:** Juanwen Yu, Chunqi Jiang, Alfabetian Harjuno Condro Haditomo, Sayaka Mino, Yuichi Sakai, Tomoo Sawabe.

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
