## [Decision Letter · Decision Letter 0]

15 Feb 2022

PONE-D-21-39007Aliamphritea hakodatensis gen. nov., sp. nov., and reclassification of Amphritea spongicola and Amphritea ceti as Aliamphritea spongicola comb. nov. and Aliamphritea ceti comb. nov. by genome taxonomyPLOS ONE

Dear Dr. Sawabe,

Thank you for submitting your manuscript to PLOS ONE. After careful consideration, we feel that it has merit but does not fully meet PLOS ONE’s publication criteria as it currently stands. Therefore, we invite you to submit a revised version of the manuscript that addresses the points raised during the review process. The reviewers raised some concerns related with the rigor of the species comparison and are requiring some extra analysis, including genomic comparisons. Please try to answer to all the reviewer's concerns.

We look forward to receiving your revised manuscript.

Kind regards,

Ivone Vaz-Moreira, PhD

Academic Editor

PLOS ONE

Journal Requirements:

“This study was supported by Kaken 19K22262.”

“NO authors have competing interests.”

4. Please take this opportunity to be sure you have met all of our guidelines for new species. When publishing papers that describe a new fungal taxon name, PLOS aims to comply with the requirements of the International Code of Nomenclature for algae, fungi, and plants (ICN). The following guidelines for publication in an online-only journal have been agreed such that any scientific fungal name published by us is considered effectively published under the rules of the Code. Please note that these guidelines differ from those for zoological nomenclature.

Effective January 2012, "the description or diagnosis required for valid publication of the name of a new taxon" can be in either Latin or English. This does not affect the requirements for scientific names, which are still to be Latin.

Also effective January 2012, the electronic PDF represents a published work according to the ICN for algae, fungi, and plants. Therefore the new names contained in the electronic publication of a PLOS ONE article are effectively published under that Code from the electronic edition alone, so there is no longer any need to provide printed copies.

For proper registration of the new taxon, we require two specific statements to be included in your manuscript.

     a.    In the Results section, the globally unique identifier (GUID), currently in the form of a Life Science Identifier (LSID), should be listed under the new species name, for example:

Hymenogaster huthii. Stielow et al. 2010, sp. nov. [urn:lsid:indexfungorum.org:names:518624]

You will need to contact either Mycobank or Index Fungorum to obtain the GUID (LSID).

    b.     In the Methods section, include a sub-section called "Nomenclature" using the following wording (this example is for taxon names submitted to MycoBank; please substitute appropriately if you have submitted to Index Fungorum and use the prefix http://www.indexfungorum.org/Names/NamesRecord.asp?RecordID= ):

The electronic version of this article in Portable Document Format (PDF) in a work with an ISSN or ISBN will represent a published work according to the International Code of Nomenclature for algae, fungi, and plants, and hence the new names contained in the electronic publication of a PLOS ONE article are effectively published under that Code from the electronic edition alone, so there is no longer any need to provide printed copies.

In addition, new names contained in this work have been submitted to MycoBank from where they will be made available to the Global Names Index. The unique MycoBank number can be resolved and the associated information viewed through any standard web browser by appending the MycoBank number contained in this publication to the prefix http://www.mycobank.org/MB/. The online version of this work is archived and available from the following digital repositories: [INSERT NAMES OF DIGITAL REPOSITORIES WHERE ACCEPTED MANUSCRIPT WILL BE SUBMITTED (PubMed Central, LOCKSS etc)].

All PLOS ONE articles are deposited in PubMed Central and LOCKSS. If your institute, or those of your co-authors, has its own repository, we recommend that you also deposit the published online article there and include the name in your article.

A complete explanation of our guidelines for publishing new species can be found on our website: http://www.plosone.org/static/guidelines#fungal

Special Cases – Algae, plant fossils, etc.

Please take this opportunity to be sure you have met all of our guidelines for new species. For submissions describing new species that do not have formal registries, please include a sub-section called “Nomenclature” in the Methods section using the following wording:

The electronic version of this article in Portable Document Format (PDF) in a work with an ISSN or ISBN will represent a published work according to the International Code of Nomenclature for algae, fungi, and plants, and hence the new names contained in the electronic publication of a PLOS ONE article are effectively published under that Code from the electronic edition alone, so there is no longer any need to provide printed copies.

The online version of this work is archived and available from the following digital repositories: PubMed Central, LOCKSS [author to insert names of any additional repositories where the work will be deposited].

Reviewers' comments:

Reviewer's Responses to Questions

**Comments to the Author**

1. Is the manuscript technically sound, and do the data support the conclusions?

Reviewer #1: Yes

Reviewer #2: Partly

2. Has the statistical analysis been performed appropriately and rigorously? 

Reviewer #1: Yes

Reviewer #2: N/A

3. Have the authors made all data underlying the findings in their manuscript fully available?

Reviewer #1: Yes

Reviewer #2: Yes

4. Is the manuscript presented in an intelligible fashion and written in standard English?

Reviewer #1: Yes

Reviewer #2: Yes

5. Review Comments to the Author

Reviewer #1: Major comments

The paper by Yamano et al describes a microbial isolate and reclassfication as a new genus and new species of the family Oceanospirillaceae. Classification using the genome as a definition of a new genus is good. However, since there is no chemical classification information, it is necessary to analyze the bacterial cell fatty acids of the isolates and compare them with Aliamphritea ceti and Aliamphritea spongicola.

Minor comments

L36: “sp., nov.” -> “sp. nov.”

L80: What does the abbreviation “ASW” mean?

L95: Delete this word "(BD)"

L111: What does the abbreviation “DDH” mean?

L119: These gene (recA, mreB, rpoA, and topA) different from the abstract is described.

L141: JCM1668T -> JCM 16668T

L163: JAMM 1525T -> JCM 14781T

L178: Please change the following reference "(Lee et al., 2018)" to a number.

L184: The minimum and maximum values in Fig. 2 are different.

L217-238: I don't know where the explanations in Fig6, Fig.7, and Table2 are separated.

Reviewer #2: The authors described novel genus by dividing the genus Amphritea based on genomic classification criteria and suggested a novel species in the novel genus. The suggestions looks reliable but required something more for demonstration. Sometimes English grammar has problems. Details are as follows.

1. About the Title and basic suggestion

A. The type species of the novel genus is demonstrated as Aliamphritea ceti, hence, Aliamphritea hakodatensis could not described as gen. nov. sp. nov. but sp. nov. only.

B. The title could be changed as “Aliamphritea gen. nov. by genome based reclassification of the genus Amphritea and reclassification of Amphritea spongicola and Amphritea ceti as Aliamphritea spongicola comb. nov. and Aliamphritea ceti comb. nov. and Aliamphritea hakodatensis sp. nov.” or “Aliamphritea ceti gen. nov. comb. nov. and Aliamphritea spongicola comb. nov. via genome based reclassification of the genus Amphritea and Aliamphritea hakodatensis sp. nov.”

C. This is same for conclusion part.

2. About the description part

A. Description of the genus looks not fully adequate. In case of temperature, described as “mesotrophic” seems better and pH and salt response, basic components of the fatty acids or polar lipids, and respiratory quinone types could be included but DNA hydrolysis looks not. Additionally, parallel description of the isolation sources could be summarized as “marine animals” or describing as “marine animal related sources such as~” seems better.

B. Does there no changes in the description of two combined species compared to former description? Please check carefully and change.

3. About the genomic features : The authors described about PHB synthesis related genes in the core genes in two genera but nothing about function of core genes in each genus. Difference of genomic features could be a reason for dividing and important in the recognition of ecological role of two genera. Please including extensive analysis on genomic features of the two genera.

4. Others

A. Figure title seems not enough for fully understanding. Additional description or legend seems required.

i. Fig. 1 : Please include compared base no. and including descriptions on size marker and bootstrap values in the figure title.

ii. Fig. 2 : Actually it’s not very important and it could be combined with Fig. 3. Move to supplementary materials or remove but including values in the Fig. 3. And ANI values against type species could be included in the Table 1.

iii. Fig. 3 : As mentioned above, ANI values could be included in the half of the table format.

iv. Fig. 5 : Require explanation on color level.

v. Fig. 6 : Considering the (C) and (D), (E) could be looks like as (A) or (B). More explanation is required.

B. About Tables

i. Table 1 could be moved to supplementary materials.

ii. Table 2 : Please include other species also and explain “I” is what meaning?

iii. Additionally, genus level comparison is required at least genera included in the Fig. 4 for genus suggestion.

C. Minors

i. Line 26: among -> above

ii. Line 28 : Who’s AAI values? It’s not clearly described.

iii. Lines 89-90: What kind of method was used for distance calculation? It should be included with options. In this term, Lines 156-158 should be moved to “materials and method” part.

iv. Line 156: What is the K2+G+I model? Please described fully.

v. Line 185: Please provided original article as reference.

6. PLOS authors have the option to publish the peer review history of their article (what does this mean?). If published, this will include your full peer review and any attached files.

Reviewer #1: No

Reviewer #2: No

---

## [Author Response · Author response to Decision Letter 0]

2 Apr 2022

Dear Professor Vaz-Moreira, academic editor, and reviewers,

We appreciate editor and reviewers for constructive suggestions. We improved the manuscript PONE-D-21-39007 according to the reviewers’ comments. Responses for specific comments are described as follows. All changes were highlighted in yellow. Also, we noticed that tables were in graphic object form, so we changed them into editable objects.

Please see point-by-point response letter file uploaded in the last.

---

## [Decision Letter · Decision Letter 1]

26 Apr 2022

PONE-D-21-39007R1Aliamphritea gen. nov. by genome taxonomy of the genus Amphritea and reclassification of Amphritea spongicola and Amphritea ceti as Aliamphritea spongicola comb. nov. and Aliamphritea ceti comb. nov. and Aliamphritea hakodatensis sp. nov.PLOS ONE

Dear Dr. Sawabe,

Thank you for submitting your manuscript to PLOS ONE. After careful consideration, we feel that it has merit but does not fully meet PLOS ONE’s publication criteria as it currently stands. Therefore, we invite you to submit a revised version of the manuscript that addresses the points raised during the review process.

We look forward to receiving your revised manuscript.

Kind regards,

Vyacheslav Yurchenko, Ph.D.

Academic Editor

PLOS ONE

Additional Editor Comments (if provided):

The work of Yamano et al. was reviewed by 4 independent reviewers. All of them have indicated further modifications to the manuscript, which needs to be introduced in order to clarify several issues. I request the authors to address them all. The revised manuscript will be peer-reviewed once again. Importantly, please do not forget to upload supplementary materials this time.

Reviewers' comments:

Reviewer's Responses to Questions

**Comments to the Author**

1. If the authors have adequately addressed your comments raised in a previous round of review and you feel that this manuscript is now acceptable for publication, you may indicate that here to bypass the “Comments to the Author” section, enter your conflict of interest statement in the “Confidential to Editor” section, and submit your "Accept" recommendation.

Reviewer #1: All comments have been addressed

Reviewer #2: (No Response)

Reviewer #3: (No Response)

Reviewer #4: (No Response)

2. Is the manuscript technically sound, and do the data support the conclusions?

Reviewer #1: Yes

Reviewer #2: Partly

Reviewer #3: Partly

Reviewer #4: Yes

3. Has the statistical analysis been performed appropriately and rigorously? 

Reviewer #1: Yes

Reviewer #2: N/A

Reviewer #3: N/A

Reviewer #4: N/A

4. Have the authors made all data underlying the findings in their manuscript fully available?

Reviewer #1: Yes

Reviewer #2: Yes

Reviewer #3: No

Reviewer #4: No

5. Is the manuscript presented in an intelligible fashion and written in standard English?

Reviewer #1: No

Reviewer #2: No

Reviewer #3: No

Reviewer #4: Yes

6. Review Comments to the Author

Reviewer #1: Major comment

The idea of "in silico chemotxonomy" is necessary as a new chemotxonomy, and the chemotxonomic information in this manuscript is described. However, it is necessary to discuss whether it can be used in a wide range of bacterial species.

I can't confirm because the supplementaly figure and table were not found.

Minor comment

P15-P24. There are many misspellings, so please reconfirm.

Table 1-3. Is it possible to summarize the order of species with Aliamphritea and Amphritea? Table 1-3 wants the order of the genera to be the same

Figures 6, 7, 8 and 9 should be moved to supplimentaly

Reviewer #2: The authors revised many but not enough for solving matters raised in previous review process. Details are as follows.

1. About the suggestion

A. As recommended, title was changed but same matters are remained in the abstract and main body of the MS.

i. Lines 34-41 also should be changed in accordance with the title. It can be suggested as follows; Depending on genome-based approaches, Aliamphritea gen. nov. was proposed with reclassification of the genus Amphritea and Aliamphritea ceti comb. nov. RA1T (=KCTC 42154T =NBRC 110551T), Aliamphritea spongicola comb. nov MEBiC05461T (=KCCM 42943T =JCM 16668T), and Aliamphritea hakodatensis sp. nov. PT3T(= JCM 34607T=KCTC 82591T). were suggested.

ii. Lines 58-63 : Please include the contents about dividing genus Amphritea into two genera.

iii. Conclusion part also should be changed.

B. Description part

i. Line 445 : remove “comb. nov.”

ii. Line 447 : remove “gen. nov.”

iii. Lines 450-451 and 457-458 : Genome of these species were newly reported, hence, including genome accession number is required.

iv. Line 482 : Provided number was not accession, please correcting it.

2. About the analysis of genome repertories

A. This MS pursuing reclassification of the genus Amphritea into two genera, hence, we will expecting the difference of genome repertories between two genera.

B. In this sense, lines 202-237 were good. And lines 243-250 also enough but Fig. 5 could be moved to supplementary materials.

C. However, lines 256-385 seems not important part of the analysis of genome repertories, these informs could be obtained by chemical analysis but ecological or physiological importance were unclear.

D. Hence, shorten the contents in lines 256-385 and strengthen the contents like in lines 202-237 (difference between two genera) seems better for readers.

3. About phenotypic comparison

A. Authors provided species by species comparison table, however, comparison at genus level is more important. Hence, Table S7 should be included in main body and Table 3 could be reduced within the range of genus Aliamphritea.

4. Others

A. Figures could be moved to supplementary or modified

i. Figs 5-6 : Could be moved to supplementary.

ii. Figs 7-9 : Moved to supplementary or simplified by bar plot type.

iii. Fig. 10 : Not very important and could be moved to supplementary.

B. About Tables

i. Table 1 : If not including that of strain PT3T it also moved to supplementary materials.

ii. Table 3 : Only members in Aliamphritea is enough.

C. Minors

i. Line 47: genera -> genus

ii. Lines 47-50 : Amphritea could be abbreviated as A. (eg A. japonica).

iii. Lines 72, 97, etc. : missing ‘,’.

iv. Lines 85 etc. : Check all PT3T in the MS, many times missing T and many times has interval between 3 and T.

v. Lines 179-182: “~ against above species~” seems enough.

Reviewer #3: Yamano et al. presented a manuscript regarding taxonomic revision of bacterial genus Amphritea. Establishing of the new genus Aliamphritea is justified by genomic analyses, which authors supplemented by chemotaxonomic analyses. Although this was a revised version, I still see a space for significant improvement, and several comments must be clarified before publication.

Major points:

I understand that the manuscript title was suggested by a reviewer, but I find it very long with unnecessary details. I suggest shortening it, e.g., “Taxonomic revision of the genus Amphritea supported by genomic and chemotaxonomic analyses”.

My major problem is lack of discussion. Although the chapter is named “Results and Discussion”, a large part of it is merely descriptive with very few biological inferences drawn from the results.

I find the usage of strain names an overkill. Surely, strain names should be clearly stated in methods, but to use them every time the species are mentioned is a bit excessive. Also, genus names should be abbreviated when used repeatedly.

Authors identified 1,660 gene clusters comprising 13,930 genes named “core”. They describe functions of very few of them on L208-211. Can authors elaborate on the rest of them?

Is difference of 10% in lipid content a significant change (Table 1)? Can those numbers be directly compared if they come from different studies? Did those studies use the same methodology? It is beyond my power to check methods in all mentioned references.

L146: Usage of chromosome 1 is not a “complete genomic sequence” as stated in the first part of the sentence. Please correct. Why did authors use only 1 chromosome of A. japonica?

Call out for Fig. 3 appears only in the legend of Fig. 2, but nowhere else in the manuscript. Comment on this figure in the main text. Moreover, I think Fig. 2 can be moved to supplement.

In Fig. 3 genes specific for a clade are called “Aliamphritea” and “Amphritea clade specific”, but in the text it is “Ceti” and “Atlantica clade specific”. Ceti and Atlantica also appears in Fig. S3. Please unify.

I would appreciate a composite figure of all synthetic pathways, i.e., adding Fig. S2 and S6 as panels to Fig. 6, and using the color codes for presence of genes in the two lineages (as in Fig. S2 now). If this was done, Table 2 could be moved to supplement.

I also do not see a point in showing genes for fatty acid, phospholipids and ubiquinone syntheses split into separate Fig. 7-9. Please combine them (using different colors for genes of different pathway). Right now, it seems that some genes from different pathway are overlapping (which may not be wrong).

In all tables, use species names instead of numbers and arrange species in a logical order (Aliamphritea species together, and Amphritea together).

L481-482: Accession numbers are not accessible. Also add corresponding accessions to supplementary tables even if they come from this study.

I think tables S3 and S4 can be merged into one listing which species were used for which analyses. Why not all species used for MLSA were used also for AAI calculations?

Fig. S1 in its current form is rather confusing. I have no idea how the numbers correspond to the species names. Can this be presented as a table?

Fig. S2 and S4 are not mentioned in the manuscript text. Please add call outs at appropriate places.

Although I consider typos and different formatting a minor thing, because of their extensive appearance throughout the whole text, I have decided to raise it as a major point. To name a few:

- L45: Oceanosprillales

- L97: missing comma

- L120 and L125: Ocenospirillaceae

- L176: missing genus names for A. opalenscence

- L224-225 and 419-421: incomplete sentences

- L273 and Table 1: Inconsistent usage of e.g. C16:0 as subscript with other parts of the text.

- Fig. 6: cis-3-decanoyl-ACP instead of cis-3-decenoyl-ACP

- L286-287: The last sentence of the paragraph is rather confusing. Rephase: “No genes responsible for the synthesis of ornithine-containing lipids were found in PT3 or any other Amphritea spp.”

Please read the whole manuscript carefully and fix typos, inconsistences, formatting, etc.

Minor points:

Abbreviations ANI, DDH, and AAI are not explained in the abstract. Please add their explanation.

Methods lack primer sequences that were used for PCR and/or sequencing.

Results and Discussion would hugely benefit from splitting the text to subchapters and adding appropriate titles.

L109-110: Include accession numbers for retrieved genome from NCBI.

L148-149: I believe that authors searched for enzymes of fatty acid, polar lipid, and isoprenoid quinone synthesis. The title should be corrected.

L152: What does “were mined from the genome sequences” mean? Proper description of tools and their parameters are necessary. Was BLAST or HMMER used? What version of the program? What was the Evalue cutoff threshold?

L185: If I correctly understand Fig. 2, the 93.7% corresponds to comparison of PT3 and A. spongicola, while 86.9% to PT3 and A. ceti. It is switched in the text.

L231: Similarly to one of my previous comments, genes do not encode putrescine. Please rephrase.

L295: Similarly, why is Phyre2 not mentioned in methods?

L267 and 311: Did authors mean linear FA by “straight-chained”?

L302: “previous chemotaxonomic properties” – Did author mean their results or previous studies? In any case, reference should be added.

L315: What is meant by “homogenous to”? Is it meant that genes of the core set in PT3 are homologous to other mentioned species?

Improve visualization of Fig. 5, right now it appears fuzzy.

What is “OF test” in Table 3? Also, “N” and “O” from the same row are not explained.

Reviewer #4: The collective Ryota Yamano et al. uses different approaches to establish the new genus Aliamphritea. The genomic data presented by authors are conclusive and original; however, before accepting the article, I suggest making a few changes.

Comments:

1. The chemotaxonomic characteristics summarized in Table 1 represents published data and is therefore not original, so I propose to move it to supplementary data.

2. Table 1: marking the species with names instead of numbers would make the table more readable.

3. I suggest improving the discussion, which is not visible between the results.

4. Unify the naming of the new species. It is recognized as PT3 in Fig 4., and Fig.7; Aliamphritea hakodatensis in Fig.3, and both names are used in Fig. 1 and Fig. 2.

5. Line 267: straight-chain change to linear-chain.

6. Fig S2. Predicted polyamine metabolism pathways in Aliamphritea and Amphritea: change to “Predicted polyamine metabolism pathways in Aliamphritea and Amphritea species.”

7. Fig S6. Predicted Q-8 synthetic pathway in Amphritea and Aliamphritea: change to “Predicted Q-8 synthetic pathway in Amphritea and Aliamphritea species. “

8. Accession numbers of genes originated from this study should be added into all tables instead of “in this study”.

9. Table S2. Genomic properties of Aliamphritea and Amphritea species: see previous point. Change Alaimphritea to Aliamphritea in the first two rows.

10. Table S3. List of genomes used for AAI calculation: see the point 8.

11. Table S7. Phenotypic characteristics of Aliamphritea and genus level comparison within the family Oceanospirillaceae: Oeanospirillum change to Oceanospirillum

Before publishing, I recommend another round of thorough review to minimize typographical errors.

7. PLOS authors have the option to publish the peer review history of their article (what does this mean?). If published, this will include your full peer review and any attached files.

Reviewer #1: No

Reviewer #2: No

Reviewer #3: No

Reviewer #4: No

---

## [Author Response · Author response to Decision Letter 1]

9 May 2022

Dear Professor Yurchenko, the academic editor, and reviewers,

We appreciate editor and reviewers for constructive suggestions. We improved the manuscript according to the reviewers’ comments. Responses for specific comments are described as follows. All changes were highlighted in yellow. 

Please see details in the point-by-point response letter uploaded separately.

best

Tomoo Sawabe

---

## [Decision Letter · Decision Letter 2]

30 May 2022

PONE-D-21-39007R2Taxonomic revision of the genus Amphritea supported by genomic and in silico chemotaxonomic analyses, and the proposal of Aliamphritea as a gen. nov.PLOS ONE

Dear Dr. Sawabe,

Thank you for submitting your manuscript to PLOS ONE. After careful consideration, we feel that it has merit but does not fully meet PLOS ONE’s publication criteria as it currently stands. Therefore, we invite you to submit a revised version of the manuscript that addresses the points raised during the review process.

ACADEMIC EDITOR: see my comments below. 

We look forward to receiving your revised manuscript.

Kind regards,

Vyacheslav Yurchenko, Ph.D.

Academic Editor

PLOS ONE

Additional Editor Comments (if provided):

The manuscript was reviewed by 2 independent referees and they both recommended further improvements. I side with all their critique and request another round of major revision. Restructuring and straightening the manuscript also seem like good ideas to me. Please note the manuscript will be peer-reviewed again.

Reviewers' comments:

Reviewer's Responses to Questions

**Comments to the Author**

1. If the authors have adequately addressed your comments raised in a previous round of review and you feel that this manuscript is now acceptable for publication, you may indicate that here to bypass the “Comments to the Author” section, enter your conflict of interest statement in the “Confidential to Editor” section, and submit your "Accept" recommendation.

Reviewer #2: All comments have been addressed

Reviewer #3: (No Response)

2. Is the manuscript technically sound, and do the data support the conclusions?

Reviewer #2: Partly

Reviewer #3: Yes

3. Has the statistical analysis been performed appropriately and rigorously? 

Reviewer #2: N/A

Reviewer #3: N/A

4. Have the authors made all data underlying the findings in their manuscript fully available?

Reviewer #2: Yes

Reviewer #3: No

5. Is the manuscript presented in an intelligible fashion and written in standard English?

Reviewer #2: No

Reviewer #3: No

6. Review Comments to the Author

Reviewer #2: The authors revised many and now could comments more about taxonomic points. In addition, English should be improved. Contents on genomic repertories also described clearly by separating common, original, and ceti group. Details are as follows.

1. Suggestion on MS structure and order of the contents in the “Results and Discussion” part

A. Divide the results into two parts, one for basic reclassification and another for further works on genomic contents.

i. In the MS, Aliamphritea mentioned before suggestion of the novel genus, it makes confusion. Hence, suggestion of the genus Aliamphritea based on phylogenetic and OGRIs results conducted first. Then, describe genome based analysis results.

ii. After suggestion of the genus name, ‘Aliamphritea Core” or “Amphritea Core” makes no problems. If not, presented as “Aliamphritea core -> ceti group core & Amphritea core -> atlantica group core” is adequate.

iii. Additionally, “Core” including Amphritea and Aliamphritea should be changed to “Common”.

B. Range of comparison in the “Phenotypic Characterization”

i. If suggesting novel genus at the first half of the Results part, phenotypic comparison between species could be reduced to species in the novel genus. As it say, Table 2 composed with first 3 species is enough. Table of this time could be given as supplementary. Of course, discussion part (lines 451-466) should be amended.

ii. Additional requirements for suggestion of novel genus: For suggestion of novel genus, genus level comparison should be given. In this case, including all genera in the family is normal process. However, Oceanospirillaceae is too big, hence, at least, compare with phylogenetically close genera such as Amphritea, Corallomonas, Neptuniibacter, Neptunomonas, Pontibaculum, Profundimonas, and Oceanispirillum is recommended. In this case, table includes basic features described in the genus is enough.

C. Possible combining or addition

i. Table S1 and S2 could be combined with all information. These two are actually overlap but inform given each is not enough.

ii. Fig. 5 : For demonstrate author’s suggestion, more plots including intra groups (atlantica vs. opalescens, japonica vs. balenae) should be given. Additionally, if evolution by inversion is proved, Fig. S3 should be moved to main text and provided by combined with Fig. 5.

iii. Fig. 6 : Genes present or not depend on species or group should be given. At present general pathway only.

2. Others

A. Require amendments

i. 16S rRNA gene based phylogeny (Fig. 1): Trimming inform and number of compared base should be given.

ii. Line 192: Need some discussion about this. Why ceti group has bigger genome?

iii. Title of Fig. 3: Of course, gene sequences from genome, it’s not genome tree. Hence, line 218 is not adequate.

B. Minors

i. Lines 26 and 120: (in silico DDH) -> (isDDH)

ii. Lines 33-34: (PT3T, A. ceti RA1T and A. spongicola MEBiC05461T) is enough

iii. Lines 37-39: change like as this “~ sp. nov. (type strain RA1T =KCTC ~)

iv. Line 106: manufactureR

v. Line 114: this is not actual accession number but “assembly accession” number. Pls change.

vi. Line 120: Average “NUCLEOTIDE” Identity

vii. Line 357: Des4 is not common, hence, need to rephrase.

viii. Line 482: 42943T), RESPECTIVELY.

ix. Line 482: novel genus -> genus Aliamphritea

x. Line 483: remove “gen. nov.”

xi. Line 498: Remove strain name, species name is enough in here.

xii. Lines 501 & 511: remove “,” after “et al.”

xiii. Line 521: insert “and: after “cream”

Reviewer #3: The authors addressed majority of the reviewers’ points, but unfortunately not all of them:

- I suggested to remove strain names from the manuscript (except for Methods). Authors indeed did so but starting by subchapter “Pan and core genome analysis”, previous subchapters still contain strain names.

- Me and reviewer #2 suggested to abbreviate species names (e.g., Amphritea to A. and Aliamphritea to Al.), but authors did not comply, because it could be confused with sea cucumber Apostichopus japonicus. I still think that abbreviating bacterial names, while keeping sea cucumber in full, would increase clarity of the text.

- One of the sentences that was incomplete remained incomplete (L464-466 now).

- Species names were not added to the tables in main text. Tables would be much clearer with them instead of numbers.

Altogether, further improvements are needed before manuscript can be considered for publication. I also raise new points that should be addressed.

New comments:

- Unify writing of thousands (with or without comma) throughout the text.

- L97: rephrase -> “were calculated using the K2 model in MEGAX as well.”

- L130: Table S2 is mentioned before Table S1 (L192), please correct.

- L162-164: rephrase -> “3D-structure of FA desaturase encoding genes from some

of the strains was predicted by Phyre2 [27].”

- L176: typo “linages” -> “lineages”

- L180: correct “which IS frequently used”

- L198: What is “Formula2”?

- L198-199: Where are DDH values shown? Is there some figure or table? Refer to it please.

- L207: According to Fig. 2, AAI value for PT3 and A. atlantica is 64.6%, so this number should be shown in the range.

- L228-229: “3-hydrokyalkanoate” -> “3-hydroXyalkanoate”

- L264: Can authors unify the use of “-aminobutyrate” in text and “4-aminobutanoate” in Fig. S2?

- L276: Call out of Fig. 4 is not appropriate here, there are no particular genes shown in this figure.

- L320 and L323: I believe it should be “comprising” or “making up” instead of “consisting”.

- L359-360: “found from” -> “found in”

- L381: Please check call outs for figure and table here, it does not look correct.

- L404-405: “which is responsible for octaprenyl-diphosphate synthesis” repeats information from two lines above.

- L406: missing comma after ubiJ

- L406: These essential genes are not present in the scheme in Fig. S9. Can authors describe their function at least in the text?

- L411: Fig. S8 is mentioned after Figs. S9 (L401) and S10 (L408), please correct.

- L423: belonged -> belonging

- L438: “PT3Twas” – missing space

- L440, 443, Table 2: Unify “Tween80”, “tween 80”, “Tween 80”.

- L447: I believe that authors wanted to say “diagnostic features […] differing from other genera were found”. If not, rewrite to make it clear.

- L484-485: Move accession numbers to Methods, there is no need for them in Conclusions.

- Fig. 3: Remove underscores from and add italics to species names.

- Fig. S3: Text of Amphritea species is fuzzy and underlined. Please correct.

- Fig. S4, S7, and S8: Can authors improve the quality of these figures?

- Table 2: *1 and *2 should be presented as superscripts.

- Table S3: *1 and *2 are not present in the table, but are explained below the table. Please, correct.

- Table S4: Typos in species names A. balenae and A. japonica.

- Table S5: Typo in Al. ceti.

7. PLOS authors have the option to publish the peer review history of their article (what does this mean?). If published, this will include your full peer review and any attached files.

Reviewer #2: No

Reviewer #3: No

---

## [Author Response · Author response to Decision Letter 2]

14 Jun 2022

Dear editor and reviewers,

We appreciate editor and reviewers for constructive suggestions. We improved the manuscript according to the reviewers’ comments. Responses for specific comments are described in a separate file. All changes were highlighted in yellow.

---

## [Decision Letter · Decision Letter 3]

27 Jun 2022

Taxonomic revision of the genus Amphritea supported by genomic and in silico chemotaxonomic analyses, and the proposal of Aliamphritea as a gen. nov.

PONE-D-21-39007R3

Dear Dr. Sawabe,

We’re pleased to inform you that your manuscript has been judged scientifically suitable for publication and will be formally accepted for publication once it meets all outstanding technical requirements.

Kind regards,

Vyacheslav Yurchenko, Ph.D.

Academic Editor

PLOS ONE

Additional Editor Comments (optional):

Reviewers' comments:

Reviewer's Responses to Questions

**Comments to the Author**

1. If the authors have adequately addressed your comments raised in a previous round of review and you feel that this manuscript is now acceptable for publication, you may indicate that here to bypass the “Comments to the Author” section, enter your conflict of interest statement in the “Confidential to Editor” section, and submit your "Accept" recommendation.

Reviewer #4: All comments have been addressed

2. Is the manuscript technically sound, and do the data support the conclusions?

Reviewer #4: Yes

3. Has the statistical analysis been performed appropriately and rigorously? 

Reviewer #4: I Don't Know

4. Have the authors made all data underlying the findings in their manuscript fully available?

Reviewer #4: (No Response)

5. Is the manuscript presented in an intelligible fashion and written in standard English?

Reviewer #4: Yes

6. Review Comments to the Author

Reviewer #4: (No Response)

7. PLOS authors have the option to publish the peer review history of their article (what does this mean?). If published, this will include your full peer review and any attached files.

Reviewer #4: No

---

## [Editor Report · Acceptance letter]

19 Jul 2022

PONE-D-21-39007R3 

Taxonomic revision of the genus *Amphritea* supported by genomic and *in silico* chemotaxonomic analyses, and the proposal of Aliamphritea gen. nov. 

Dear Dr. Sawabe:

I'm pleased to inform you that your manuscript has been deemed suitable for publication in PLOS ONE. Congratulations! Your manuscript is now with our production department. 

Kind regards, 

on behalf of

Prof. Vyacheslav Yurchenko 

Academic Editor

PLOS ONE